EMBO
**Molecular Medicine**

# Dantrolene corrects cellular disease features of Darier disease and may be a novel treatment

Matthew Hunt [ID][1,8], Nuoqi Wang[1,8], Naricha Pupinyo[1], Philip Curman[1,2,3], Monica Torres [ID][1,2], William Jebril[1,2], Maria Chatzinikolaou [ID][1], Julie Lorent[4], Gilad Silberberg[5], Ritu Bansal [ID][1], Teresa Burner[6], Jing Zhou[7], Susanne Kimeswenger [ID][6], Wolfram Hoetzenecker[6], Keith Choate[7], Etty Bachar-Wikstrom [ID][1✉] & Jakob D Wikstrom [ID][1,2✉]

## Abstract

**Darier disease (DD) is a rare severe acantholytic skin disease caused by mutations in the *ATP2A2* gene that encodes for the sarco/endoplasmic reticulum calcium ATPase isoform 2 (SERCA2). SERCA2 maintains endoplasmic reticulum calcium homeostasis by pumping calcium into the ER, critical for regulating cellular calcium dynamics and cellular function. To date, there is no treatment that specifically targets the disease mechanisms in DD. Dantrolene sodium (Dl) is a ryanodine receptor antagonist that inhibits calcium release from ER to increase ER calcium levels and is currently used for non-dermatological indications. In this study, we first identified dysregulated genes and molecular pathways in DD patient skin, demonstrating downregulation of cell adhesion and calcium homeostasis pathways, as well as upregulation of ER stress and apoptosis. We then show in various in vitro models of DD and SERCA2 inhibition that Dl aided in the retention of ER calcium and promoted cell adhesion. In addition, Dl treatment reduced ER stress and suppressed apoptosis. Our findings suggest that Dl specifically targets pathogenic mechanisms of DD and may be a potential treatment.**

**Keywords** Darier Disease; Dantrolene; ER Calcium; Cell Adhesion; UPR
**Subject Categories** Genetics, Gene Therapy & Genetic Disease; Skin

## Introduction

Darier disease (DD), is a rare cutaneous disease inherited in an autosomal dominant mode. Recent reports estimate the prevalence of DD to be around 1 in 55,000 worldwide (Yeshurun et al, 2021) and 1 in 12,000 in Sweden (Zagoras et al, 2023). Classic characteristics include scaly, crusted papules and plaques in seborrheic areas and flexures, primarily caused by the loss of intercellular adhesion and disordered keratinization (Cooper and Burge, 2003). Aside from symptom management and behavioral modifications to avoid triggers, there are currently no validated treatments for DD. A variety of treatments have been proposed including retinoids, steroids, vitamin D analogs, photodynamic therapy, and surgical excision. However, the beneficial effects are modest, adverse side effects are frequent and limit their use, and relapse commonly occurs following treatment discontinuation (Klausegger et al, 2013).

The underlying defect of DD pathogenesis is the result of heterozygous mutations in the *ATP2A2* gene which encodes sarco/endoplasmic reticulum calcium ATPase isoform 2 (SERCA2). By catalyzing the hydrolysis of adenosine triphosphate (ATP), SERCA2 pumps two $Ca^{2+}$ ions from the cytosol into the lumen of endoplasmic reticulum (ER), thus maintaining a higher $Ca^{2+}$ concentration in the ER compared to the cytosol (Lytton et al, 1992). Mutations in *SERCA2* lead to a loss of $Ca^{2+}$ transport, subsequently decreasing $Ca^{2+}$ transport into the ER and a depletion of ER-$Ca^{2+}$ stores (Bachar-Wikström and Wikström, 2021). ER-$Ca^{2+}$ store depletion triggers an ER stress response (Savignac et al, 2014), and persistent ER stress in DD keratinocytes can lead to the upregulation of apoptosis (Rogers et al, 1990). In addition, impaired desmoplakin trafficking is associated with the loss of cell-to-cell adhesion (Mauro, 2014), which in homeostatic settings is mediated by $Ca^{2+}$-dependent interactions between the extracellular domains of desmosomal cadherins (Dhitavat et al, 2004).

Dantrolene (Dl) is a ryanodine receptor (RyR) inhibitor which stabilizes the resting state of the channel—preventing the release of $Ca^{2+}$ from intact ER to help maintain ER-$Ca^{2+}$ stores (Choi et al, 2017). Notably, Dl is used to relax muscles in malignant hyperthermia syndrome (Kim et al, 2019), as well as for the treatment of cardiac arrythmias (Gramley et al, 2009). More recently, Dl has also been reported to have neuroprotective effects against neuroleptic malignant syndrome (Simon et al, 2022),

[1]Dermatology and Venereology Division, Department of Medicine (Solna), Karolinska Institutet, Stockholm, Sweden. [2]Dermato-Venereology Clinic, Karolinska University Hospital, Stockholm, Sweden. [3]Department of Medical Epidemiology and Biostatistics (Solna), Karolinska Institutet, Stockholm, Sweden. [4]National Bioinformatics Infrastructure Sweden (NBIS), Science for Life Laboratory, Department of Biochemistry and Biophysics, Stockholm University, Stockholm, Sweden. [5]Bioinformatics & Computational Biology Research Operations, Champions Oncology Inc, Rehovot, Israel. [6]Johannes Kepler University Linz, Kepler University Hospital Linz, Department of Dermatology, Linz, Austria. [7]Department of Dermatology, Genetics, and Pathology, Yale University School of Medicine, New Haven, CT, USA. [8]These authors contributed equally: Matthew Hunt, Nuoqi Wang. ✉E-mail: ester.bachar-wikstrom@ki.se; jakob.wikstrom@ki.se

spasticity, heat stroke, and ecstasy intoxication (Inan and Wei, 2010). As such, we decided to investigate whether Dl is beneficial in the context of DD pathophysiology and in particular, to restoring ER-$Ca^{2+}$ homeostasis and subsequently alleviating the pathogenic mechanisms underpinning DD pathology.

## Results and discussion

### Gene expression profiling of lesional and non-lesional DD skin

Using full-thickness biopsies of both lesional (L) and non-lesional (NL) sections of skin derived from DD patients (Fig. 1A), we first sought to investigate differences in gene expression through RNAseq gene expression analysis. Differential expression gene (DEG) analysis determined there to be 3044 significantly upregulated genes in L skin compared to NL skin, while there were 2443 genes significantly downregulated in L vs NL skin (Fig. 1B).

Principal component analysis (PCA) demonstrated that the mRNA expression profiles were separated by whether the samples were from L or NL skin (Fig. EV1A). Cell-type deconvolution was able to detect signatures of multiple skin cells, including keratinocytes and endothelial cells. Fibroblasts, however, were not detected in any of the biopsies (Fig. EV1B; Appendix Fig. S1). To assess whether major cell types differ in their abundance between tissue types, we compared the scores of endothelial cells and keratinocytes between lesional and non-lesional samples (Appendix Fig. S2), and found no significant differences in abundances, indicating that gene expression differences are not attributable to differences in tissue composition (Fig. EV1B).

GO enrichment analysis demonstrated that upregulated genes in L skin were involved in pathways including DNA post-processing, whilst the top downregulated genes were associated with mitochondrial processes and metabolism (Fig. EV1C). As far as we are aware, this is this is the first observation of the role of mitochondria in DD pathophysiology (Bachar-Wikström and Wikström, 2021). Due to the significance of ER-mitochondria contact sites in $Ca^{2+}$ signaling (Marchi et al, 2018), and of mitochondria in other skin-related pathologies (Hunt et al, 2023), our results suggest a potentially important role of mitochondria in DD pathophysiology.

Of note, we identified several pathways known to be associated with DD pathophysiology that were significantly upregulated in L skin. These included those of ER stress, $Ca^{2+}$ homeostasis, apoptosis, and cell adhesion regulation (Fig. 1C). This confirmed previous studies on DD pathophysiology, including in confirming how differences in SERCA2 levels between lesional and non-lesional skin impacts DD pathogenesis (Celli et al, 2011, 2012; Dhitavat et al, 2003; Hobbs et al, 2011; Savignac et al, 2014; Wang et al, 2011). Subsequently, in this study we then sought to assess the effect of Dl on these pathways in various in vitro models of DD or SERCA2 inhibition.

### Dantrolene improves ER-$Ca^{2+}$ homeostasis

As abnormal $Ca^{2+}$ homeostasis within the ER is one of the hallmarks of DD, we investigated the impact of Dl treatment on preventing the loss of ER-$Ca^{2+}$ stores. First, using either Human epidermal keratinocyte (HEKa) cells treated with the 10 nM

thapsigardin (Tg)—a SERCA2 inhibitor, or primary DD patient keratinocytes (DDK) stained with Fura Red AM, we found that Dl treatment significantly reduced baseline as well as peak levels of cytoplasmic $Ca^{2+}$ following the addition of 1 µM Tg in both DDK (Fig. 1D,E), and Tg HEKa cells (Figs. 1F,G and EV2A,B).

To investigate ER-$Ca^{2+}$, cells were transfected with an ER-specific $Ca^{2+}$-sensitive GFP reporter (ER-GCaMP6-150 GFP). Here, Dl pre-treatment significantly increased baseline ER-$Ca^{2+}$ in both DDK (Figs. 1H,I and EV2C,D) and HEKa (Fig. 1J,K) cells, and significantly inhibited $Ca^{2+}$ efflux following the addition of 1 µm Tg. Dl treatment also significantly helped retain ER-$Ca^{2+}$ levels in fibroblasts isolated from both L and NL biopsies (Fig. EV2E,F). Collectively, these finding corresponds to the advantageous effect of Dl on ER-$Ca^{2+}$ levels seen in other diseases (Gramley et al, 2009; Inan and Wei, 2010; Riazi et al, 2018).

Due to the fact that there is an increase in NFAT transcription factor translocation in response to elevated intracellular $Ca^{2+}$ levels (Maguire et al, 2013), we sought to determine whether Dl treatment decreased the expression of various NFAT-related genes. Here, Dl treatment significantly decreased the expression of NFAT1 and NFAT4 in Tg-treated HEKa cells (Fig. EV3A), with a trend in Dl-treated immortalized DDK (Fig. EV3B), confirming that increases in cytosolic $Ca^{2+}$ induced NFAT pathway activation (Zhou et al, 2021). As STIM-ORAI $Ca^{2+}$ entry machinery is commonly activated in DD keratinocytes in response to the loss of ER-$Ca^{2+}$ (Maguire et al, 2013; Stanisz et al, 2022), future studies should investigate the potentially beneficial effect of Dl on its activity. Finally, as mitochondria act as important regulators in $Ca^{2+}$ homeostasis (Giorgi et al, 2021), the impact of Dl on mitochondrial $Ca^{2+}$ would additionally be an interesting avenue to pursue in future studies.

### Dantrolene promotes cell adhesion

We next sought to analyze cell adhesion in various in vitro models. Firstly, the gene expression of DSG2, DSG3, E-CAD, DSP, DSC3, and CLDN4 was significantly increased in immortalized DDK/Tg+ vs DDK/Tg− cells (Fig. 2A).

Next, through immunofluorescence analysis of, we found that Dl treatment significantly improved the expression of important cell adhesion and junction markers, namely β-catenin, occludin (OCLN), and desmoglein-1 (DSG-1) in primary DDK/Dl+ vs DDK/Dl− cells (Fig. 2B,C). Expression of all three markers was also significantly higher in Tg+/Dl+ vs Tg+/Dl− cells (Fig. 2D,E), as well as in siATP2A2/Dl+ vs siATP2A2/Dl− cells (Fig. 2F,G). Immunoblot analysis of SERCA2 confirmed the downregulation of ATP2A2 in siATP2A2 HEKa cells (Fig. EV4A,B).

Finally, we investigated cell adhesion in the HEKa models of SERCA2 inhibition through a dispase mechanical assay. In the Tg HEKa model, fragment number was significantly lower, and fragment size was significantly greater at both 10 nM and 20 nM Tg in Dl+ vs Dl− cells (Fig. 2H,I). Similarly, Dl treatment also significantly decreased fragment number and significantly increased fragment size in siATP2A2/Dl+ vs siATP2A2/Dl− cells (Fig. 2J,K).

As desmosomal-regulated cell adhesion has been shown to be modulated by $Ca^{2+}$ homeostasis in normal skin epidermis (Duden and Franke, 1988; Hobbs et al, 2011; Pillai et al, 1988), as well as in

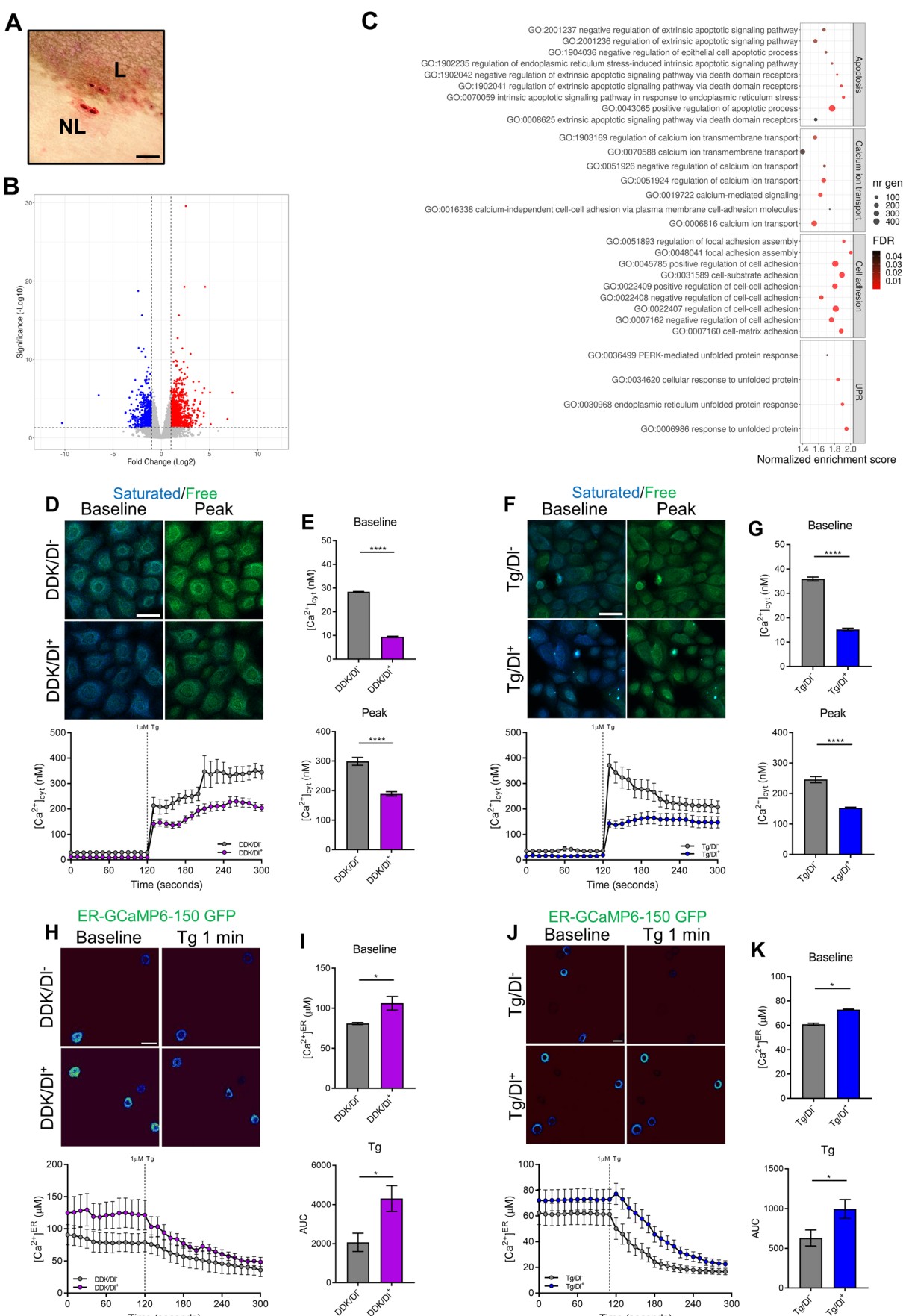

**Figure 1. RNAseq demonstrates impaired Ca²⁺ homeostasis, cell adhesion, and ER stress in DD.**

(A) Representative image of lesional (L) and non-lesional (NL) biopsies taken from DD patients. Scale bar = 10 mm. (B) Volcano plot of differentially expressed genes in L skin vs NL skin. Genes significantly upregulated in L skin (red), and significantly downregulated (blue). Dotted lines indicate the significance threshold for -Log$_{10}$ significance and Log$_2$ fold change. P values were adjusted by the Benjamini–Hochberg method. (C) Gene Ontology analysis of significantly upregulated pathways related to apoptosis, Ca²⁺ transport, cell adhesion, and the UPR in L vs NL skin. Dots are plotted by normalized enrichment score, colored by the False Discovery Rate (FDR), and sized depending on the number of expressed genes in each pathway. (D, F) Representative images and graph (mean ± SEM) of Fura Red AM staining and cytosolic Ca²⁺ concentration in (D) primary DKK and (F) Tg-treated HEKa. Color channel is adjusted in the same manner. N = 3 biological repeats containing at least 15 cells in each condition. Scale bars = 25 μm. (E, G) Graphs (mean ± SEM) of baseline Ca²⁺ and peak Ca²⁺ concentration following 1 μM Tg addition in (E) primary DKK and (G) Tg-treated HEKa. Two-way ANOVA; ****$P \leq 0.0001$. Color channel is adjusted in the same manner. (H–K) Graphs (mean ± SEM) and pseudocolored representative images of ER-Ca²⁺ imaging in (H) primary DDK and (J) HEKa cells. The color channel is adjusted in the same manner. Scale bar = 25 μm. N = 3 biological repeats in each condition. Source data are available online for this figure.

keratinocytes treated with SERCA2 inhibitors (Dhitavat et al, 2003; Savignac et al, 2014), our results suggest that the improved Ca²⁺ homeostasis induced by Dl treatment has downstream effects on cell adhesion in DDK and keratinocyte models of SERCA2 inhibition. This would support previous findings whereby miglustat, a molecular chaperone, rescued cell adhesion impairment induced by ER stress in DD keratinocytes (Savignac et al, 2014).

## Altered UPR in response to dantrolene treatment

As upregulation of the unfolder protein response (UPR) is a common characteristic of ER-Ca²⁺ dyshomeostasis and ER stress in general, whilst DD-derived keratinocytes have been shown to exhibit hallmarks of constitutive ER stress (Savignac et al, 2014), we tested whether Dl ameliorated ER stress through RT-qPCR of ER stress genes. First, when comparing immortalized DDK cells to immortalized control keratinocytes, we found that the expression of *CHOP, ERO1, ERN1, GRP78, ATF6*, and spliced *XBP1* (*s/usXBP1*) were all significantly greater in the DDK cells. Although not significant, there was a trend of reduced expression in the respective markers when comparing DDK/Dl⁺ to DDK/Dl⁻ cells (Fig. 3A).

With regards to the other keratinocyte models, *CHOP* was significantly downregulated in Tg⁺/DL⁺ vs Tg⁺/Dl⁻ cells, however, no other markers were significantly changed (Fig. 3B). In addition, *ERO1* and *ERN1* expression was significantly reduced in siNEG/Dl⁺ vs siNEG/Dl⁻ cells (Fig. 3C). Dl treatment also significantly reduced *ATF6* expression in siATP2A2/Dl⁺ cells, while *s/usXBP1* was significantly upregulated with Dl treatment (Fig. 3C). Finally, all UPR markers were significantly downregulated in NL/Dl⁺ DDF vs NL/Dl⁻ DDF, whilst *CHOP, ERN1, GRP78*, and *ATF6* were significantly downregulated in L/Dl⁺ DDF vs L/Dl⁻ DDF (Fig. 3D).

Taken together, these results suggest that Dl ameliorated ER stress and the UPR. Due to the fact that ER are responsible for the regulation of the epidermal Ca²⁺ gradient (Lee et al, 1992; Menon et al, 1985), and that ER stress and the UPR is upregulated in the event of calcium dyshomeostasis (Bahar et al, 2016), the beneficial effect of Dl treatment on the UPR is most likely due to the restoration of ER-Ca²⁺ homeostasis (Onozuka et al, 2006). As the UPR is known to induce inflammatory responses in the skin (Meares et al, 2014), it would be interesting to investigate the effect of Dl on these inflammatory pathways, such as the JAK-STAT pathway, in future studies. Indeed, previous anecdotal observations suggested a beneficial effect of JAK inhibitors on DD (Aihie and Dyer, 2023).

## Dantrolene prevents caspase 3/7 apoptosis

Next, we investigated the effect of Dl treatment on cell survival as well as the rate of apoptosis induction. Here, Dl treatment had no significant effect on cell viability in immortalized DDK/Dl⁺ vs immortalized DDK/Dl⁻ cells (Fig. 3E). Interestingly, Dl treatment significantly increased cell viability in Tg⁺/Dl⁺ HEKa vs Tg⁺/Dl⁻ cells at 10 nM Tg (Fig. 3F). However, there was no significant difference in cell viability in either siATP2A2 HEKa (Fig. 3G), or DDF cells (Fig. 3H).

Finally, we assessed whether Dl treatment normalized the rate of apoptosis. Here, Dl treatment significantly reduced caspase 3/7 activity in DDK/Dl⁺ vs DDK/Dl⁻ cells (Fig. 3I), as well as in Tg⁺/Dl⁺ vs Tg⁺/Dl⁻ HEKa cells (Fig. 3J). However, there was no significant difference in siATP2A2/Dl⁺ vs siATP2A2/Dl⁻ cells (Fig. 3K). In DDF, although no significant difference in apoptosis was seen between NL/Dl⁻ and L/Dl⁻ cells, Dl treatment significantly reduced the rate of apoptosis in L/Dl⁺ and NL/Dl⁺ compared to L/Dl⁻ and NL/Dl⁻ cells respectively (Fig. 3L). Overall, our results suggest that Dl promoted cell viability and inhibited apoptosis in vitro. Apoptosis has been shown to be increased in DD pathophysiology (Onozuka et al, 2006; Savignac et al, 2014). In particular, rounded keratinocytes (corps round), which are a hallmark of DD, are thought to arise from uncontrolled UPR and ER stress, subsequently leading to apoptosis (Godic, 2004; Macleod and Munro, 1991). As apoptosis is induced when the UPR fails to resolve ER stress (Corazzari et al, 2017), our results suggest that the beneficial effect of Dl on decreasing the rate of the UPR, and in particular the expression of *CHOP* (Hu et al, 2019), may subsequently prevent the induction of apoptosis (Bachar-Wikström and Wikström, 2021; Hu et al, 2019).

## Conclusion

In this study, we confirmed pathophysiological differences between lesional and non-lesional skin of DD patients through RNAseq analysis and subsequently tested the therapeutic potential of Dl on alleviating these pathophysiological factors in in vitro models of DD and SERCA2 inhibition.

Collectively, we found that Dl helped retain ER-Ca²⁺ levels, subsequently preventing cell adhesion dysregulation, as well as induction of the UPR and apoptosis. Overall, these findings highlight the potential therapeutic benefits of Dl in treating DD pathophysiology. Although current treatments have been reported to be efficacious - primarily retinoids - low to non-existent long-term patient acceptability exists due to various adverse effects (Burge, 1999; Burge and Wilkinson, 1992; Engin et al, 2015). Interestingly, recent studies have demonstrated the potential

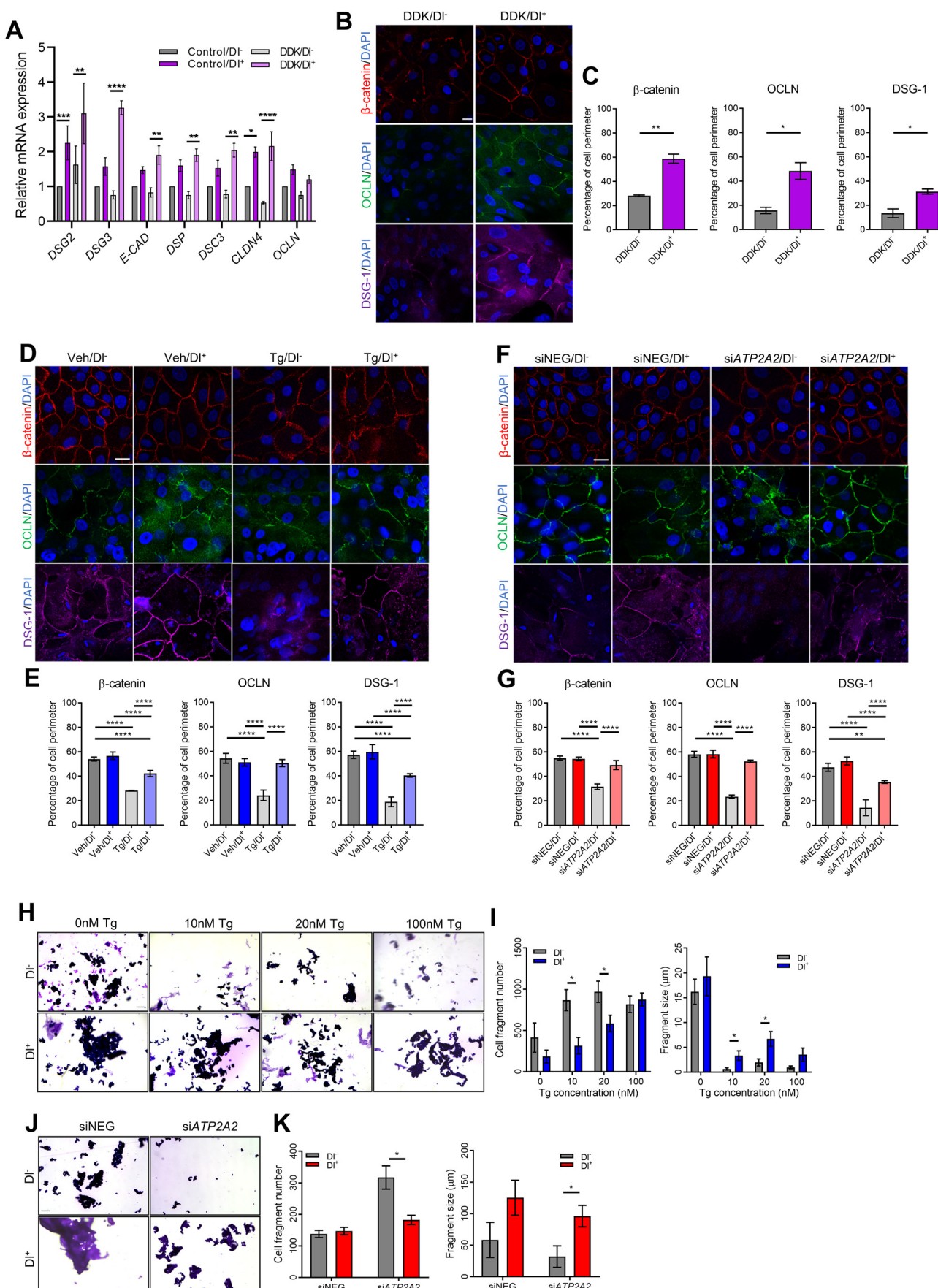

**Figure 2. Dantrolene promotes cell adhesion.**

(A) RT-qPCR analysis (mean + SEM) of cell adhesion genes after normalization to *ACTB* in immortalized DDK. $n = 3$ biological replicates containing three technical replicates. Two-way ANOVA; ****$P \le 0.0001$; ***$P \le 0.001$; **$P \le 0.01$; *$P \le 0.05$. (B, D, F) Representative fluorescence images of cell adhesion markers β-catenin, OCLN, and DSG-1 in (B) primary DDK, (D) Tg HEKa, and (F) si*ATP2A2* HEKa cells. Colors are adjusted in the same manner for Dl⁻ and Dl⁺ images. Scale bar = 20 μm. (C, E, G) Bar graphs (mean + SEM) of the percentage of cell perimeter covered by the respective cell adhesion markers in (C) primary DDK, (E) Tg HEKa, and (G) si*ATP2A2* HEKa cells. Paired *t* test and two-way ANOVA; **$P \le 0.01$; *$P \le 0.05$. $N = 10$ cells in three biological replicates. (H, J) Representative brightfield images of the dispase mechanical assay in (H) Tg HEKa and (J) si*ATP2A2* HEKa models. Scale bar = 500 μm. (I, K) Cell fragment number and fragment size (μm) graphs (mean + SEM) for (I) Tg HEKa, and (K) si*ATP2A2* HEKa models. Two-tailed *t* test. *$P \le 0.05$. $N = 3$ biological replicates with at least 30 fields of view. Source data are available online for this figure.

applicability of targeting skin inflammation in DD, in particular the IL-17/IL-23 axis (Amar et al, 2023; Ettinger et al, 2023; Haghighi Javid et al, 2023). In another recent report, inhibition of the MEK pathway mitigated adhesion and differentiation defects in organoid models of DD (Zaver et al, 2023). As no animal models currently exist for DD, and that Dl is a safe and already approved drug, we suggest the next logical step to be a clinical trial.

# Methods

## Ethics

This study was approved by the Stockholm Regional Ethics Committee and executed in agreement with the Helsinki Declaration and the Department of Health and Human Services Belmont Report. Informed consent was obtained from all the research subjects.

## Darier disease patient samples

In total, 12 DD research subjects were enrolled at the Karolinska University Hospital Dermatology Clinic, Stockholm, Sweden. In each patient, two samples were harvested from both lesional and adjacent non-lesional skin with a 4-mm biopsy punch. The samples used for RNAseq were snap frozen in liquid nitrogen and the samples used for fibroblast isolation were put in culture media. Patients consented to sharing of pathology photography.

## RNA-sequencing gene expression analysis

Total RNA was extracted from 4-mm full-thickness punch biopsies from both lesional and non-lesional DD patients using the RNeasy Mini Kit (Qiagen). RNA quality was determined by both a Qubit 3.0 (Thermo Fisher Scientific) and Bioanalyzer 2100 (Agilent). Library preparation was performed with the Illumina TruSeq stranded mRNA kit (Illumina) from 500 ng of RNA for each sample. Libraries were normalized and pooled before the pool was measured by qPCR. Sequencing was performed on the NovaSeq 6000 Sequencing System (Illumina) using NovaSeq 6000 v1.5 reagents (Illumina). Reads were quality-checked and aligned to reference genomes using an established pipeline (https://nf-co.re/rnaseq) (Patel et al, 2023). After quality control check of the genes (Anders and Wolfgang, 2010), and exclusion of genes with low expression (less than 3 samples with at least 10 counts), differently expressed genes were identified using DESeq2 version 1.38.0 (Love et al, 2014), with *P* values adjusted by the Benjamini–Hochberg method and a False Discovery Rate (FDR) threshold of 0.05. Gene

ontology (GO) enrichment analysis then performed on org.Hs.eg.db Bioconductor package version 3.16.0 and Enrichr (https://maayanlab.cloud/Enrichr/), with only GO terms classed as Biological Processes considered. Enrichment *P* values were also adjusted using the Benjamini–Hochberg method. Cell-type deconvolution was performed on variance stabilizing transformed (VST) read counts using xCell (Aran et al, 2017). Thirty-eight cell types expected to be present in the biopsies were included (Fig. EV1B; Appendix Fig. S2).

## Cell culture and treatments

In this study, we used various different in vitro models for DD or SERCA2 inhibition. First, we utilized primary and immortalized keratinocytes isolated from DD patients (DDK), as well as primary fibroblasts (DDF) isolated from both lesional and non-lesional DD skin. All of which had mutations in *ATP2A2*. In addition, we standardized primary keratinocyte (HEKa) models with either pharmacological or genetic SERCA2 inhibition. For pharmacological inhibition, HEKa cells were treated with thapsigargin (Tg), a chemical inhibitor of SERCA pumps. For genetic inhibition, we employed the use of HEKa cells transfected with siRNA against *ATP2A2* (si*ATP2A2*).

DDK isolated from DD patients with confirmed mutations in *ATP2A2*, their corresponding immortalized version, as well as primary and immortalized keratinocytes from a healthy control (Kepler University Hospital Linz and Yale University School of Medicine respectively) were cultured in keratinocyte growth media 3 (PromoCell) supplemented with 0.06 mM CaCl₂, Supplement mix (PromoCell), and antibiotics (100 units/mL Penicillin and 100 μg/mL Streptomycin) (Thermo Fisher Scientific). HEKa (Thermo Fisher Scientific) were cultured in EpiLife with 60 μM calcium, serum free keratinocyte growth media supplemented with human keratinocyte growth supplement (HKGS) and antibiotics (100 units/mL Penicillin and 100 μg/mL Streptomycin). DDF were isolated from DD patients (both lesional and non-lesional tissue) and cultured in DMEM glucose free media (Thermo Fischer Scientific) supplemented with 10% fetal bovine serum (FBS), 10 mM glucose, and antibiotics (100 units/mL penicillin and 100 μg/mL streptomycin). All cell lines were maintained in an incubator set at 37 °C and 5% CO₂.

Cells were treated with 10 μM Dl (Sigma-Aldrich) and either 10 nM or 20 nM Tg (Sigma-Aldrich) for 24 h, where stated. For the dispase mechanical assay, cell viability, and apoptosis assays, Tg was used at concentrations of 10, 20, and 50 nM for 24 h, as stated. For RT-qPCR analysis of NFAT markers, HEKa cells were treated with the NFAT inhibitor VIVIT (5 μM) (Bio-Techne) for 24 h as a positive control.

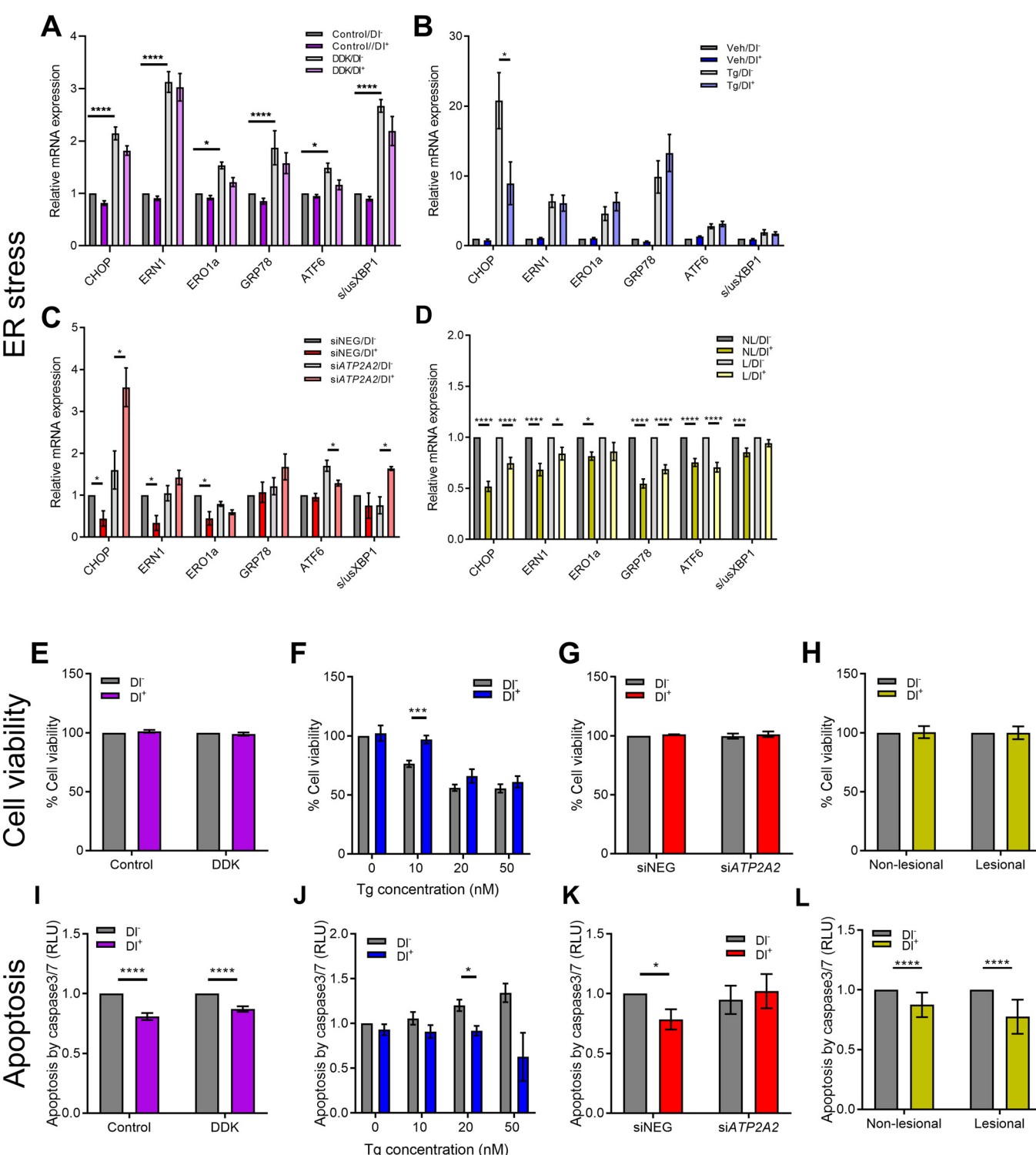

## Transfections

*ATP2A2* knockdown was performed in HEKa cells using 2.5 pmol siRNA targeted to *ATP2A2* (si*ATP2A2*) (Thermo Fisher Scientific) for 48 h with Lipofectamine™ 2000 (Thermo Fisher Scientific). Simultaneously, universal negative (Sigma-Aldrich) was used for 24 h to create siNEG cells.

For the fluorescent measurement of ER-Ca²⁺, primary DDK, Tg and si*ATP2A2* HEKa cells, as well as DDF, cells were transfected with a Ca²⁺-GFP reporter localized to ER (CMV ER-GCaMP6-150, from Addgene (plasmid #86918) for 24 (keratinocytes), or 48 (fibroblasts) hours using Lipofectamine™ 3000 (Thermo Fisher Scientific).

◄ **Figure 3. Dantrolene prevents the unfolded protein response and apoptosis.**

(A–D) RT-qPCR analysis (mean + SEM) of UPR markers in (A) immortalized DDK, (B) Tg HEKa, (C) si*ATP2A2* HEKa, (C) DDF models. All genes are normalized to *ACTB* gene expression. Two-way ANOVA; ****$P \leq 0.0001$; ***$P \leq 0.001$; *$P \leq 0.05$; $N = 3$ biological replicates containing three technical replicates. (E–H) Graphs (mean + SEM) depicting percentage cell viability in (E) immortalized DDK, (F) Tg-treated HEKa, (G) siNEG and si*ATP2A2* HEKa, and (H) L and NL DDF, all with and without Dl treatment for 24 h. Two-way ANOVA; ***$P \leq 0.001$; $N = 3$ biological replicates containing three technical replicates. (I–L) Graphs (mean + SEM) depicting Relative Light Unit (RLU) caspase 3/7 activity assay quantification of apoptosis in (I) immortalized DDK, (J) Tg-treated HEKa, (K) siNEG and si*ATP2A2* HEKa, and (L) L and NL DDF, all with and without Dl treatment for 24 h. Two-way ANOVA; ****$P \leq 0.0001$; *$P \leq 0.05$; $N = 3$ biological replicates containing three technical replicates. Source data are available online for this figure.

## RT-qPCR

Total RNA was extracted from cells using Trizol reagent (Thermo Fisher Scientific). After determining the sample quality, reverse transcription was performed using the RevertAid First Strand cDNA Synthesis Kit (Thermo Fisher Scientific). Gene expression was quantified by SYBR Green expression assays and normalized to *ACTB*. The primer information used in these experiments are listed in Appendix Fig. S3.

## Immunoblotting

Protein was isolated from si*ATP2A2*-treated HEKa cells and then subjected to western blotting. SERCA2 protein expression was detected with anti-SERCA2 antibody (1:1000, #4388, Cell Signaling) and HRP-conjugated anti-rabbit IgG secondary antibody (1:10,000, #7074, Cell Signaling), while β-actin protein expression was detected with HRP-conjugated β-actin antibody (1:10,000, #A3854, Sigma-Aldrich).

## Ca²⁺ imaging

Both ER-Ca²⁺ and cytosolic Ca²⁺ imaging was performed on a Zeiss Airyscan LSM880 confocal microscope in an environmental chamber that maintained the temperature at 37 °C and 5% CO₂ and imaged at ×20 magnification. For DDK and Tg HEKa, cells were seeded onto μ-slide eight-well glass bottom plates (Ibidi) and either pre-treated with 20 μM Dl or vehicle for 24 h. Following time-lapse imaging of baseline levels, 1 μM Tg was added.

To calibrate ER-Ca²⁺ concentration (Eq. (1)), we first used 4 μM ionomycin (Thermo Scientific Fisher) in Ca²⁺ and Mg²⁺-free HBSS to get the minimum fluorescence ($F_{min}$) and then changed the media to HBSS with 2 mM Ca²⁺, 1 mM Mg²⁺, and 25 μM digitonin (Thermo Scientific Fisher) for the maximum fluorescence ($F_{min}$) (Palmer et al, 2004).

For DDF cells, following transfection and preincubation with Tg and or Dl, an initial baseline (0 min) image was taken before cells were treated with 10 nM Tg and images over 15-min time points.

For Cytosolic Ca²⁺ imaging, DDK and HEKa were seeded onto μ-slide eight-well glass bottom plates and stained with the ratiometric dye Fura Red AM (5 μM in HBSS with 2 mM Ca²⁺ (Thermo Scientific Fisher) + 0.05% Pluronic acid F-127 (Sigma-Aldrich) for 1 h followed by a 30-min rest at 37 °C). Time-lapse imaging was performed at ×20 magnification using the 488 nm laser with 420 nm and 480 nm excitation wavelengths and 576–638 nm and 638–759 nm emission filters, with images taken at 10 s intervals. DDK and HEKa cells were pre-treated with 10 nM Tg (Tg HEKa model) and either 20 μM Dl or vehicle for 24 h.

Following the setting of imaging conditions based on unstained cells and calcium concentration calibration (Eq. (2)) with 100 μM BAPTA-AM (*Rmax*) (R&D Systems) and 4 μM ionomycin (*Rmin*), time-lapse imaging of baseline levels was followed by addition of 1 μM Tg.

$$[Ca^{2+}]_{ER} = K_d[(F_{ER-Ca}/F_{max} - 1/R_f)/1 - F_{ER-Ca}/F_{max}]^{1/n} \quad (1)$$

Equation (1)—Kd is the affinity constant of the indicator, $F_{ER-Ca}$ is the measured fluorescence, Rf is the dynamic range (Fsat/Fapo) and n is the Hill coefficient.

$$[Ca^{2+}]_{cyt} = Kd\,Q[(R - R\min)/(R\min - R)] \quad (2)$$

Equation (2)—where R represents the fluorescence intensity ratio Fλ1/Fλ2, in which λ1 and λ2 are the fluorescence detection wavelengths for the ion-bound and ion-free indicator, respectively. Kd is the Ca²⁺ dissociation constant of the indicator. Q is the ratio of Fmin to Fmax at λ2.

## Immunofluorescence staining and imaging of cell adhesion markers

DDK, as well as Tg- and si*ATP2A2*-treated HEKa cells, were seeded onto μ-slide eight-well glass bottom plates (Ibidi) and treated with 20 μM Dl for 24 h, followed by treatment with 2 mM calcium chloride (Thermo Fisher Scientific) as follows: 24 h for β-catenin (1:500, #8814, Cell Signaling Technology), 240 h for OCLN (1:100, ab216327, Abcam), and 288 h for DSG-1 (1:100, #PA5-106826, Thermo Fischer Scientific). Next, the cells were fixed with 4% paraformaldehyde (PFA) (Thermo Fisher Scientific) for 10 min then permeabilized by 0.1% tritonX-100 (Sigma-Aldrich) treatment for 10 min, followed by staining with primary antibodies against the above proteins for 1 h at RT. After a wash cycle with phosphate-buffered saline (PBS) with calcium and magnesium (Thermo Fisher Scientific), cells were incubated with Alexa Fluor donkey anti-rabbit 555 (1:1000, #A-21428, Thermo Fischer Scientific) for 1 h at RT before incubation with DAPI (1:2000) for 10 min at RT.

For fluorescence microscopy of cell adhesion markers, cells were imaged at either ×20, ×40, or ×63 magnification on a Zeiss Airyscan LSM880 using filters for Alexa 555 alongside Alexa 405 for DAPI. For analysis, in at least ten cells in each condition, the perimeter of the cells as well as the length covered by the respective cell adhesion markers was quantified and used to calculate the percentage of cell perimeter with positive staining for cell adhesion markers of interest. Images were analyzed on ImageJ software.

**The paper explained**

**Problem**

Darier disease is a severe and rare skin condition caused by inherited autosomal dominant mutations in the ER calcium pump SERCA2, which moves calcium into the ER. These mutations lead to epidermal acantholysis (loss of cell adhesion). Current treatments face challenges related to efficiency, specificity, and potential adverse effects, highlighting the need for novel therapeutic approaches. Dantrolene sodium, a drug used for non-dermatological indications, specifically inhibits the ryanodine receptor, reducing the efflux of ER calcium and potentially counteracting faulty SERCA2. Since there are no animal models for Darier disease, research must rely on cellular models to develop new treatments.

**Results**

In this study, we first demonstrate gene expression differences that underpin various pathophysiological factors of Darier disease between lesional and non-lesional patient skin samples, including calcium signaling, cell adhesion, and ER stress. Using in vitro models of Darier disease—either mutated primary patient keratinocytes or keratinocytes with pharmacological inhibition of SERCA2—we show through confocal microscopy-based fluorescence imaging that dantrolene sodium improves the retention of ER calcium and correspondingly lowers cytosolic calcium. Consequently, we demonstrate that this improved ER calcium homeostasis results in enhanced cell–cell adhesion, as evidenced by immunofluorescence imaging and an enzymatic fragmentation assay, as well as decreased ER stress and apoptosis.

**Impact**

Here, we show for the first time that dantrolene sodium improves key pathophysiological factors underpinning Darier disease pathogenesis. Dantrolene may represent a novel treatment for Darier disease and warrants further testing in clinical trials.

## Cellular viability assay

Cell viability was determined using resazurin-based PrestoBlue (Thermo Fisher Scientific), with absorbance read at both 560 and 600 nm on a Glomax multi-detection system microplate reader (Promega).

## Caspase 3/7 apoptosis assay

Measurements of caspase 3 and 7 activity as an indicator of the levels of apoptosis were performed using the Caspase-Glo 3/7 assay (Promega) according to the manufacturer's instructions. Relative luminescence units (RLU) were measured using a Glomax multi-detection system microplate reader (Promega) and used to quantify apoptotic activity.

## Dispase mechanical assay

To quantify cell adhesion and mechanical strength, cell models were subjected to a mechanical dispase dissociation assay (as described previously (Ishii et al, 2005)). Briefly, Dl-treated si*ATP2A2*, as well as Tg-treated HEKa cells, were subjected to dispase treatment (Invitrogen) and fixation with 4% PFA (Thermo Fisher Scientific). Cells were then stained with 0.0025% crystal violet (Sigma-Aldrich) and transferred onto 8 well glass slides for acquisition on a Zeiss Axio Scan Z1 digital slide scanner. Images were analyzed on ImageJ, and the

count, total area, average size, and percentage area of the cell fragments were quantified in order to report the cell fragment number as well as the fragment area (μm).

## Statistical analysis

Statistical significance was determined by paired two-tailed Student's *t* test. The significance among multiple groups was determined by one-way or two-way ANOVA by GraphPad Prism Version 6. Pearson's correlation test on log10-transformed data was performed by using GraphPad Prism Version 6, with *P* value < 0.05 determined to be statistically significant.

# Data availability

RNA-sequencing data is available in the following database: 582 https://www.ebi.ac.uk/biostudies/arrayexpress (Accession number: E-MTAB-13754).

The source data of this paper are collected in the following database record: biostudies:S-SCDT-10_1038-S44321-024-00104-3.

# Peer review information

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

## Acknowledgements

The authors thank the patients for participating in this study and providing the biopsies as well as research nurse Helena Griehsel. This work was supported by grants from Hudfonden, Swedish Science Council, Swedish Society for Medical Research, Leo Foundation, ALF medicin Stockholm, Jeanssons Stiftelse, Wallenberg Foundation and Tore Nilssons Stiftelse. The authors also acknowledge support from the National Genomics Infrastructure in Stockholm funded by Science for Life Laboratory, the Knut and Alice

Wallenberg Foundation and the Swedish Research Council, and SNIC/Uppsala Multidisciplinary Center for Advanced Computational Science for assistance with massively parallel sequencing and access to the UPPMAX computational infrastructure. Support by NBIS (National Bioinformatics Infrastructure Sweden) is gratefully acknowledged.

## Author contributions

**Matthew Hunt**: Data curation; Software; Formal analysis; Validation; Investigation; Visualization; Methodology; Writing—original draft; Writing—review and editing. **Nuoqi Wang**: Data curation; Software; Formal analysis; Validation; Investigation; Visualization; Methodology; Writing—original draft; Writing—review and editing. **Naricha Pupinyo**: Formal analysis; Validation; Investigation; Methodology; Writing—review and editing. **Philip Curman**: Formal analysis; Validation; Investigation; Methodology; Writing—review and editing. **Monica Torres**: Investigation; Visualization; Writing—review and editing. **William Jebril**: Investigation; Writing—review and editing. **Maria Chatzinikolaou**: Investigation; Writing—review and editing. **Julie Lorent**: Data curation; Formal analysis; Methodology; Writing—review and editing. **Gilad Silberberg**: Data curation; Software; Formal analysis; Validation; Writing—review and editing. **Ritu Bansal**: Methodology; Writing—review and editing. **Teresa Burner**: Resources; Writing—review and editing. **Jing Zhou**: Resources; Writing—review and editing. **Susanne Kimeswenger**: Resources; Writing—review and editing. **Wolfram Hoetzenecker**: Resources; Writing—review and editing. **Keith Choate**: Resources; Writing—review and editing. **Etty Bachar-Wikstrom**: Conceptualization; Resources; Data curation; Supervision; Funding acquisition; Validation; Investigation; Visualization; Methodology; Writing—original draft; Project administration; Writing—review and editing. **Jakob D Wikstrom**: Conceptualization; Resources; Data curation; Supervision; Funding acquisition; Validation; Investigation; Visualization; Methodology; Writing—original draft; Project administration; Writing—review and editing.

Source data underlying figure panels in this paper may have individual authorship assigned. Where available, figure panel/source data authorship is listed in the following database record: biostudies:S-SCDT-10_1038-S44321-024-00104-3.

## Funding

## Disclosure and competing interests statement

The authors declare no competing interests.

# Expanded View Figures

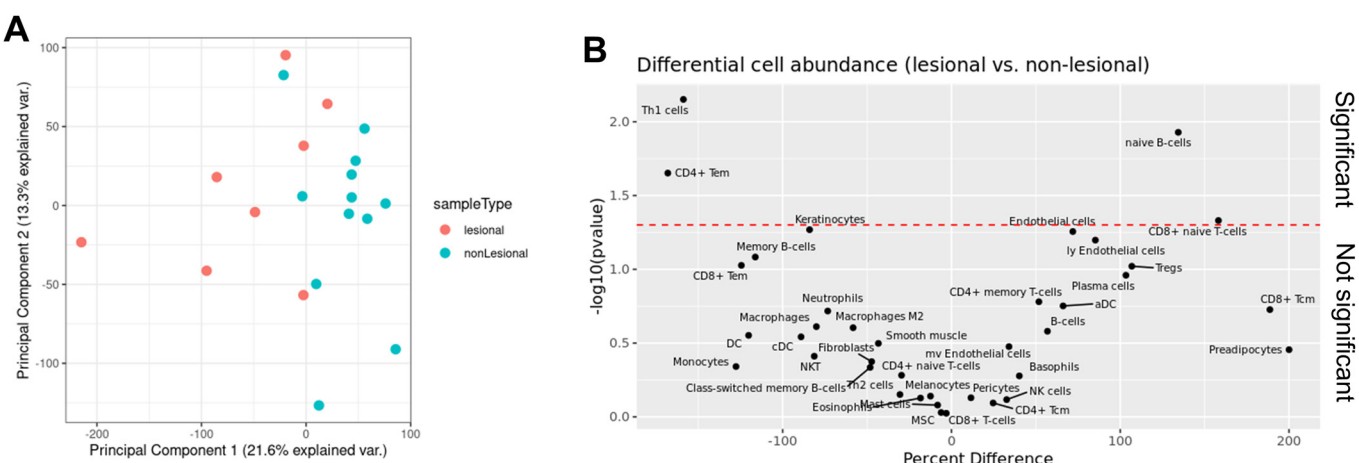

**Figure EV1. RNAseq analysis of lesional and non-lesional DD skin biopsies.**

(A) Principal component analysis (PCA) plots based on mRNA expression profiles. Each dot represents an individual patient. Red = lesional skin; blue = non-lesional skin. (B) Table of the top ten most significantly differently expressed Gene Ontology (GO) pathways in lesional vs non-lesional skin. NES = normalized enrichment score; size = number of genes in each respective pathway. (C) Volcano plot showing differential cell abundance of cell types between lesional and non-lesional biopsies derived from deconvolution analysis. Statistical significance determined through paired *t* tests.

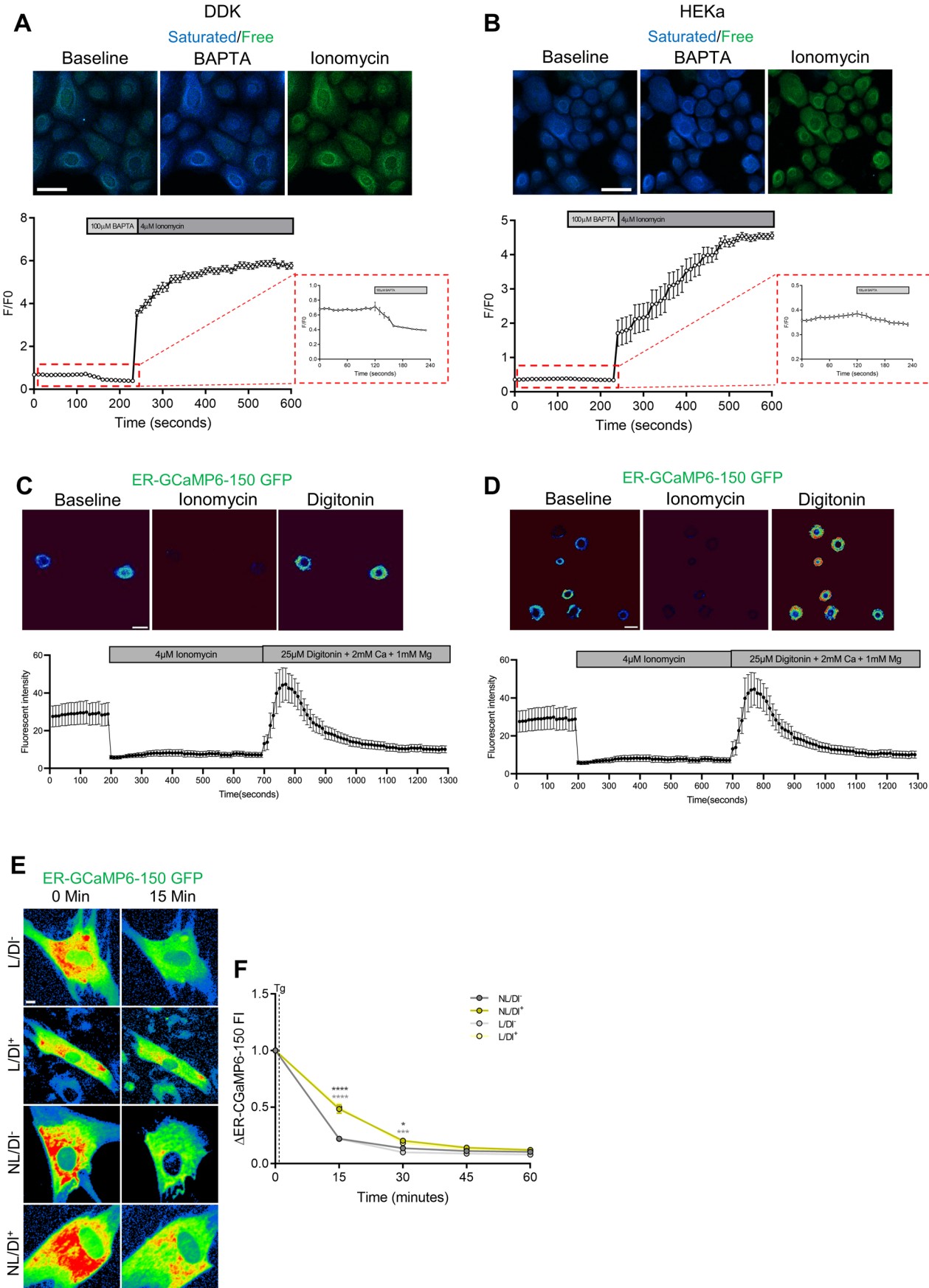

**Figure EV2.  Ca²⁺ live cell imaging calibration.**

(A, B) Graph and representative images of Fura Red cytosolic Ca²⁺ calibration in (A) DDK and (B) Tg HEKa. Scale bars = 25 µm. (C, D) Graph and representative images of ER-Ca²⁺ calibration in (C) DDK and (D) Tg HEKa. Scale bar = 25 µm. (E) Representative pseudocoloured images of ER-Ca²⁺ imaging in DDF. Colors are adjusted in the same manner. Scale bar = 10 µm. (F) Graph (mean ± SEM) of ER-Ca²⁺ fluorescence in DDF. Two-way ANOVA; ****$P < 0.0001$; ***$P \leq 0.001$; *$P \leq 0.05$. Source data are available online for this figure.

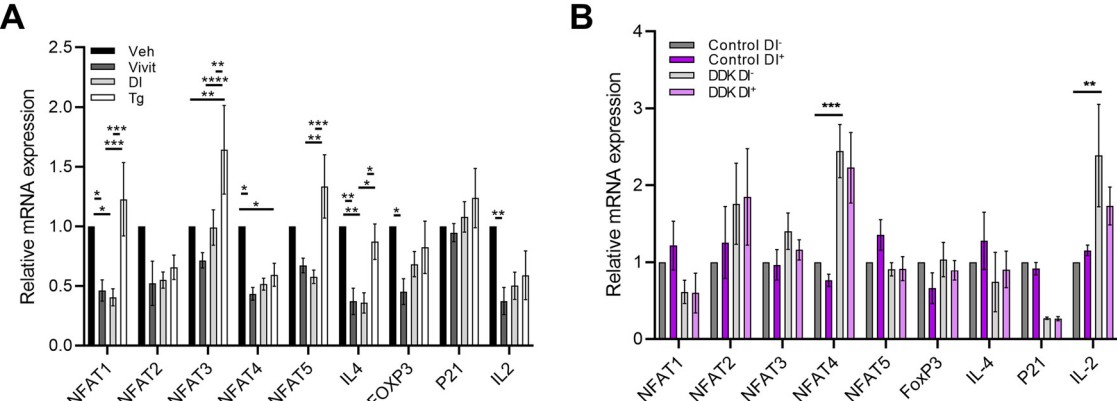

**Figure EV3. Gene expression analysis of NFAT markers.**

(A, B) RT-qPCR quantification (mean ± SEM) of relative mRNA gene expression of different *NFAT* markers in (A) HEKa cells and (B) DDK cells. Vivit was used as a negative control to inhibit *NFAT* marker expression. *P* values were calculated by two-way ANOVA; ****$P \leq 0.0001$; ***$P \leq 0.001$; **$P \leq 0.01$; *$P \leq 0.05$; $N = 3$ biological replicates containing 3 technical replicates for all conditions. Source data are available online for this figure.

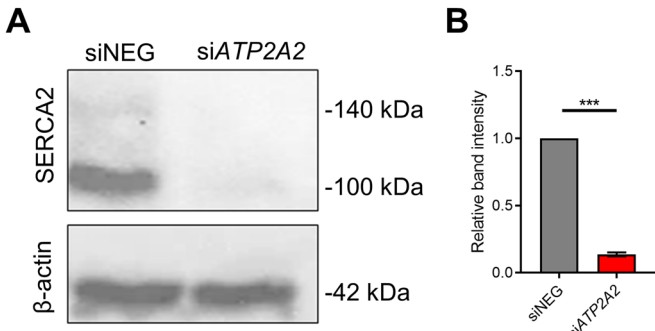

**Figure EV4.   Optimization of *ATP2A2* knockdown.**

(**A**) Representative immunoblots depicting SERCA2 downregulation in HEKa cells transfected with either siNEG or si*ATP2A2*. (**B**) Relative band intensity quantification (mean ± SEM) of SERCA2 protein expression after normalization to β-actin. *P* values two-way ANOVA; ***$P \leq 0.001$. $N = 3$ biological replicates.

