## [Peer Review File · EMBO Molecular Medicine]

Dantrolene corrects cellular disease features of Darier disease and may be a novel treatment

Matthew Hunt, Nuoqi Wang, Naricha Pupinyo, Philip Curman, Monica Torres, William Jebril, Maria Chatzinikolaou, Julie Lorent, Gilad Silberberg, Ritu Bansal, Teresa Burner, Jing Zhou, Susanne Kimeswenger, Wolfram Hötzenecker, Keith Choate, Ety Bachar-Wikstrom, and Jakob Wikstrom

Corresponding author(s): Jakob Wikstrom (jakob.wikstrom@ki.se) , Ety Bachar-Wikstrom (ester.bachar-wikstrom@ki.se)

Review Timeline:

Submission Date:	5th Jun 23
Editorial Decision:	4th Jul 23
Revision Received:	30th Jan 24
Editorial Decision:	4th Mar 24
Revision Received:	5th Mar 24
Editorial Decision:	7th Mar 24
Revision Received:	26th Jun 24
Accepted:	4th Jul 24

Editor: Zeljko Durdevic

Transaction Report:

4th Jul 2023

Dear Dr. Wikstrom,

Thank you for the submission of your manuscript to EMBO Molecular Medicine. We have now received feedback from the three reviewers who agreed to evaluate your manuscript. All three referees recognize potential interest of the study but also raise important and partially overlapping criticism that should be addressed in a major revision. For further consideration of the manuscript, it is necessary to repeat essential experiments in patient keratinocytes or to at least show that the knockdown of ATP2A2 in primary keratinocytes resembles the cellular disease pathology. Experiments using animal model of Darier disease and addition of Hailey-Hailey disease patients as suggested by the referee #1 are not required and should be addressed by discussion in the manuscript. Further, I would like you to consider reformatting your manuscript to a scientific report type of article (3 figures, ~22000 characters), for more information please check our "Author Guidelines".
<https://www.embopress.org/page/journal/17574684/authorguide#reportsarticleguide>

We would welcome the submission of a revised version within three months for further consideration. Please let us know if you require longer to complete the revision.

I look forward to receiving your revised manuscript.

Yours sincerely,

Zeljko Durdevic

We require:

- 1) A .docx formatted version of the manuscript text (including legends for main figures, EV figures and tables). Please make sure that the changes are highlighted to be clearly visible.
- 2) Individual production quality figure files as .eps, .tif, .jpg (one file per figure). For guidance, download the 'Figure Guide PDF': (<https://www.embopress.org/page/journal/17574684/authorguide#figureformat>).
- 3) A .docx formatted letter INCLUDING the reviewers' reports and your detailed point-by-point responses to their comments. As part of the EMBO Press transparent editorial process, the point-by-point response is part of the Review Process File (RPF), which will be published alongside your paper.
- 4) A complete author checklist, which you can download from our author guidelines

(<https://www.embopress.org/page/journal/17574684/authorguide#submissionofrevisions>). Please insert information in the checklist that is also reflected in the manuscript. The completed author checklist will also be part of the RPF.

6) It is mandatory to include a 'Data Availability' section after the Materials and Methods. Before submitting your revision, primary datasets produced in this study need to be deposited in an appropriate public database, and the accession numbers and database listed under 'Data Availability'. Please remember to provide a reviewer password if the datasets are not yet public (see <https://www.embopress.org/page/journal/17574684/authorguide#dataavailability>).

12) For more information: There is space at the end of each article to list relevant web links for further consultation by our readers. Could you identify some relevant ones and provide such information as well? Some examples are patient associations,

relevant databases, OMIM/proteins/genes links, author's websites, etc...

13) Author contributions: You will be asked to provide CRediT (Contributor Role Taxonomy) terms in the submission system. These replace a narrative author contribution section in the manuscript.

14) A Conflict of Interest statement should be provided in the main text.

Please note: When submitting your revision you will be prompted to enter your funding and payment information. This will allow Wiley to send you a quote for the article processing charge (APC) in case of acceptance. This quote takes into account any reduction or fee waivers that you may be eligible for. Authors do not need to pay any fees before their manuscript is accepted and transferred to the publisher.

EMBO Press participates in many Publish and Read agreements that allow authors to publish Open Access with reduced/no publication charges. Check your eligibility: <https://authorservices.wiley.com/author-resources/Journal-Authors/open-access/affiliation-policies-payments/index.html>

***** Reviewer's comments *****

Referee #1 (Comments on Novelty/Model System for Author):

In Darier disease the keratinocytes are skin cells that are mostly affected by the loss of function mutations of the SERCA pump. However, the authors selected HEK cells and for some of the experiments patient-derived fibroblasts as a working model. I do not fully understand why were keratinocytes not used at least for some of the experiments. Confirming the findings in an animal model of DD plus and minus Dantrolone would increase the clinical relevance of the study.

Since Danrelone is in clinical use, the authors might want to search patient databases for individuals treated with dantralone and evaluate how this might have affected gene expression or protein abundance in the skin or in outer organs. This might be helpful in enhancing the relevance of the mechanisms identified in HEK cells.

Referee #1 (Remarks for Author):

In this manuscript the authors propose Dantrolene sodium (D1) as potential drug in treating patients with Darier disease caused by mutations in the SERCA pump (ATP2A2).

Indeed, this is an interesting subject as novel cures for DD as well as Hailey-Hailey disease (HHD) are very much needed. However, the study needs significant improvement and additional experimental investigation in order to prove if Dantrolene can be used in treating DD and HHD.

I have several suggestions and comments:

1. Given that keratinocytes are most affected cells the authors should try to identify gene alterations only in these cells (experimentally or at least bioinformatically). Here, I refer to the experiments depicted in Fig 1 and S1. It is very likely that the gene expression in the biopsy samples are strongly affected by fibroblasts, melanocytes and even immune cells of the skin.
2. Referring to Fig. 2, additional calcium measurements are needed. Given the importance of ER Ca²⁺ signaling, it is very likely that the cytosolic as well as mitochondrial homeostasis will be affected in cells with non-functional SERCA. Also more detailed information about the sensor used to measure ER calcium is needed. For this study using a ratiometric one is essential. Normalizing the values to 1 leads to loss of a very important information i.e. the resting levels before stimulation, which are very likely also altered.
3. The study would profit if patients with HHD would be involved.
4. Regarding the study design and selection of experimental tools please see my comments in section 4.
5. In experiments in which Thapsigargin was used the authors should consider the fact that Tg induces ER stress.
6. Given that SERCA inactivity leads to ER calcium depletion, the STIM-ORAI calcium entry machinery (already indicated as prognostic markers for DD and HHD) is most likely activated in these cells (keratinocytes). This would also mean that NFAT is

more active and that many of the genes altered in the DD patients are under control of NFAT. Is Dantrolene affecting NFAT activity?

7. The mitochondrial contribution requires further examination.

Minor.

HEK cell treated with siRNA or similar should not be referred to as disease model.

the figure call-outs should be checked. For example Fig. 2G is mentioned before Fig. 2A-E.

A full list of genes from the RNAseq with the corresponding fold change and P values should be provided.

Referee #2 (Comments on Novelty/Model System for Author):

This reports repurposes dantrolene, used to treat muscle spasticity or malignant hyperthermia, to treat the impaired endoplasmic reticulum calcium storage and defective cell-to-cell adhesion associated with Darier's skin disease. The experiments are carefully done. More experimental detail could be added, as sometimes it is difficult to understand the experimental approach. In addition, the conclusion that clinical trials are the next step appears to be overstated, as dantrolene only appears to be effective in reversing functional defects only in low doses of thapsigargin. However, these criticisms should be relatively easily addressed by the authors.

Referee #2 (Remarks for Author):

General remarks: This reports repurposes dantrolene, used to treat muscle spasticity or malignant hyperthermia, to treat the impaired endoplasmic reticulum calcium storage and defective cell-to-cell adhesion associated with Darier's skin disease. The experiments are carefully done. More experimental detail could be added, as sometimes it is difficult to understand the experimental approach. In addition, the conclusion that clinical trials are the next step appears to be overstated, as dantrolene only appears to be effective in reversing functional defects only in low doses of thapsigargin.

Specific Criticisms:

Methods:

1) For genetic analysis, were whole skin samples or samples containing only epidermis analyzed. If whole skin samples, why was dermis included since the pathology appears to be in the epidermis only?

2) Cell culture: what was the calcium concentration in the cell culture medium?

3) Calcium measurements: please include what solutions were used for calcium measurements. If media was used as the solution, were the solutions gassed with appropriate CO₂?

Fig 2. Its difficult to understand exactly what happened in this figure without going back to the methods. Please add this information in the figure legend.

Figure 5. There appear to be variable responses for everything except CHOP. Explanation?

Figure 6. Dantrolene appears to be effective only at TG 10 nM dose for cell viability. Do the authors know what its effect is on ER calcium in micromolar concentrations? Figure 2 only shows a normalized fluorescence intensity.

Discussion: the authors state that there have been no clinical trials for Darier's disease. It's more accurate to say there have been no successful trials. Also its worth discussing whether the changes in adhesion and apoptosis are large enough in cell culture to warrant a clinical trial.

Referee #3 (Comments on Novelty/Model System for Author):

The authors chose to use siRNA knockdown of ATP2A2 in primary keratinocytes despite ready access to darier patients - knockdown is not the same as at the haploinsufficiency seen in DD patients. It's not clear that the model is true to the disease state. Furthermore, use of DD fibroblasts is odd given the absence of clear pathology in these cells in human disease.

Referee #3 (Remarks for Author):

The authors chose to use siRNA knockdown of ATP2A2 in primary keratinocytes despite ready access to darier patients - knockdown is not the same as at the haploinsufficiency seen in DD patients. It's not clear that the model is true to the disease state. Furthermore, use of DD fibroblasts is odd given the absence of clear pathology in these cells in human disease. This data is not a surrogate for studying patient keratinocytes.

It is not surprising that a calcium channel inhibitor mitigates the effects of decreased ER calcium stores in DD..

The authors should cite manuscripts showing the IL17 inhibition has efficacy in DDD.

Figure 2: The authors appear to be seeing calcium retention in excess of normal keratinocytes (siNEG/DL), nearly 2-fold greater. Normalization would be expected to be more relevant. Panels C and D are from non-relevant cells and have expected results. E appear to have a typo on right upper panel. While confocal is interesting - graphical presentation of analysis of multiple cells would be of greater benefit. F - it is not clear why fibroblast data is presented.

Given suggestion of therapeutic effect, have the authors considered RNA sequencing of normal keratinocytes, their atp2a2 knockdown cells, and the same cells treated with DL to show global normalization of gene function.

Overall, my greatest concern is the use of a non-physiologic model for DD. While the findings presented may be true, it is equally possible that they do not reproduce disease features of DD.

Comments from the editor:

“For further consideration of the manuscript, it is necessary to repeat essential experiments in patient keratinocytes or to at least show that the knockdown of ATP2A2 in primary keratinocytes resembles the cellular disease pathology. Experiments using animal model of Darier disease and addition of Hailey-Hailey disease patients as suggested by the referee #1 are not required and should be addressed by discussion in the manuscript.”

Answer: Firstly, we would like to thank you for considering our study for resubmission. We have, as suggested, repeated several experiments in patients' keratinocytes (as well as performed other experiments as outlined below). Regarding Hailey-Hailey we address this disease in the Discussion.

“Further, I would like you to consider reformatting your manuscript to a scientific report type of article (3 figures, ~22000 characters), for more information please check our "Author Guidelines".”

Answer: The manuscript contains a considerable amount of data, especially after all the revision experiments requested, and it is not possible for us to cut it. We hope you have understanding for this.

Reviewer's comments

Referee #1 (Comments on Novelty/Model System for Author):

In Darier disease the keratinocytes are skin cells that are mostly affected by the loss of function mutations of the SERCA pump. However, the authors selected HEK cells and for some of the experiments patient-derived fibroblasts as a working model. I do not fully understand why were keratinocytes not used at least for some of the experiments. Confirming the findings in an animal model of DD plus and minus Dantrelone would increase the clinical relevance of the study. Since Danrelone is in clinical use, the authors might want to search patient databases for individuals treated with dantralone and evaluate how this might have affected gene expression or protein abundance in the skin or in outer organs. This might be helpful in enhancing the relevance of the mechanisms identified in HEK cells.

Answer: We would like to thank the reviewer for investing time in reviewing our manuscript. With regards to the first point made, we would like to clarify that we used HEKa cells, which are primary keratinocyte cells, either with pharmacological or genetic SERCA2 inhibition. With regards to the point about animal models of DD, we did not use one as there are no validated animal models currently established which recapitulate the pathophysiology of DD (the knockout mice have a completely different phenotype, they get skin cancer instead of the typical DD rash). Instead, we have performed new experiments in patient derived keratinocytes (**Figures 2-5**). And finally, with regards to the point about patient databases, we have been unable to find any case reports about the use of DI in DD patients, and so believe there not to be any.

Referee #1 (Remarks for Author):

In this manuscript the authors propose Dantrolene sodium (D1) as potential drug in treating patients with Darier disease caused by mutations in the SERCA pump (ATP2A2).

Indeed, this is an interesting subject as novel cures for DD as well as Hailey-Hailey disease (HHD) are very much needed.

However, the study needs significant improvement and additional experimental investigation in order to prove if Dantrolene can be used in treating DD and HHD.

I have several suggestions and comments:

1. Given that keratinocytes are most affected cells the authors should try to identify gene alterations only in these cells (experimentally or at least bioinformatically). Here, I refer to the experiments depicted in Fig 1 and S1. It is very likely that the gene expression in the biopsy samples are strongly affected by fibroblasts, melanocytes and even immune cells of the skin.

Answer: In the original experiment, we tested the extraction of RNA from epidermis only, dermis only, and epidermis + dermis and found that the RIN values from dermis samples were negligible (thus indicating that little dermis was included due to careful dissection). As such, in order to avoid altering the samples by lengthy enzymatic removal of the dermis from the epidermis, we reasoned that using epidermis + dermis samples was most appropriate. With regard to the reviewer's comments, we performed deconvolution analysis of the bulk RNAseq data through the xCell tool and found that there was negligible amounts of fibroblasts in the samples used, and that the main population of cells was keratinocytes (**Figure EV 1C**). In addition, through this analysis we were also able to determine that there were very few significant differences in cell type composition between lesional and non-lesional biopsies. Regarding Hailey-Hailey disease, please see our answer to point 3 below.

2. Referring to Fig. 2, additional calcium measurements are needed. Given the importance of ER Ca²⁺ signaling, it is very likely that the cytosolic as well as mitochondrial homeostasis will be affected in cells with non-functional SERCA. Also more detailed information about the sensor used to measure ER calcium is needed. For this study using a ratiometric one is essential. Normalizing the values to 1 leads to loss of a very important information i.e. the resting levels before stimulation, which are very likely also altered.

Answer: In response to the reviewer's helpful comments, we have added a graphical representation to summarise the experimental setup (**Figure 2A**), as well as adding more detailed text about the experimental methods in a new 'Ca²⁺ imaging' section in the methods. In addition, as well as the ER-Ca²⁺ analysis, we have additionally performed experiments to analyse cytosolic calcium dynamics using the ratiometric dye Fura Red AM, as well as mitochondrial calcium through live cell imaging of Rhod-2 AM.

3. The study would profit if patients with HHD would be involved.

Answer: Whilst we agree with and appreciate the comment about the benefit of including HHD patients in the study due to the similarities between HHD and DD, we feel that studying the effect of DI or another novel treatment in the context of HHD should be performed separately, as in this study we aimed to specifically investigate the effect on SERCA2 and therefore ER, not Golgi. We plan to study HHD in the near future, but as the pathophysiology is less understood than DD at present, we felt that this was not in the scope of this manuscript.

4. Regarding the study design and selection of experimental tools please see my comments in section 4.

Answer: We hope that we have addressed this comment in the reply to point 2.

5. In experiments in which Thapsigargin was used the authors should consider the fact that Tg induces ER stress.

Answer: We appreciate the comment. Tg is a well-known inhibitor of SERCA2 and so is widely used in studies investigating DD, in which ER stress is also an acknowledged feature. As such, ER stress was an accepted consequence of using Tg in our study, and ER stress was measured in Tg-treated HEK293 cells (**Figure 4B**). In addition, we note that incubation may be an issue in experiments with longer incubation times, but in our experiments we used a low concentration of Tg and only for 24 hours. Finally, as patient keratinocytes has been included the manuscript relies less on the Tg model.

6. Given that SERCA inactivity leads to ER calcium depletion, the STIM-ORAI calcium entry machinery (already indicated as prognostic markers for DD and HHD) is most likely activated in these cells (keratinocytes). This would also mean that NFAT is more active and that many of the genes altered in the DD patients are under control of NFAT. Is Dantrolene affecting NFAT activity?

Answer: We thank the reviewer for this helpful comment. In response, we assessed the gene expression of various NFAT markers as well as downstream targets of significance through RT-qPCR and found that DI reduced the expression of *NFAT1* and *NFAT4* in HEK293 cells. In addition, we found that NFAT expression was significantly higher in mutant keratinocytes compared to control, and that there was a trend towards DI decreasing expression of various NFAT-related genes in the mutant DDK patient cells (**Figure EV 3**). These findings are mentioned in the Results as well as Discussion.

7. The mitochondrial contribution requires further examination.

Answer: We agree that the contribution of mitochondria in DD is interesting and requires further examination. However, we feel that investigation of this requires significant work and so is not in the scope of this paper. Indeed, we plan to investigate this thoroughly in a separate study which is part of PhD student project in our lab.

Minor.

HEK cell treated with siRNA or similar should not be referred to as disease model.

Answer: We agree with this comment and as such have changed all references to Tg-treated HEKa or siATP2A2 HEKa as models of SERCA2 inhibition, with primary DD keratinocytes now only being referred to as disease models.

The figure call-outs should be checked. For example Fig. 2G is mentioned before Fig. 2A-E.

Answer: We appreciate the raising of this comment and have address it throughout the manuscript.

A full list of genes from the RNAseq with the corresponding fold change and P vallues should be provided.

Answer: We agree with this comment and as such have included the full list of RNAseq genes and their differential expression analysis to the manuscript (**Appendix Figure S1**).

Referee #2 (Comments on Novelty/Model System for Author):

This reports repurposes dantrolene, used to treat muscle spasticity or malignant hyperthermia, to treat the impaired endoplasmic reticulum calcium storage and defective cell-to-cell adhesion associated with Darier's skin disease. The experiments are carefully done. More experimental detail could be added, as sometimes it is difficult to understand the experimental approach. In addition, the conclusion that clinical trials are the next step appears to be overstated, as dantrolene only appears to be effective in reversing functional defects only in low doses of thapsigargin. However, these criticisms should be relatively easily addressed by the authors.

Answer: We thank the reviewer for the possible outlook on the manuscript and for taking the time to review it. In response to the reviewer's helpful comments, we have added a graphical representation to summarise the experimental setup (**Figure 2A**), as well as adding more detailed text about the experimental methods in a new 'Ca²⁺ imaging' section in the methods. In addition, as well as the ER-Ca²⁺ analysis, we have additionally performed experiments to analyse intracellular calcium dynamics using the ratiometric dye Fura Red AM, as well as mitochondrial calcium through live cell imaging of Rhod-2 AM (**Figure 2B-G**). With regards to their comment regarding the statement about clinical trials, we hope that including additional experimental data using primary DD keratinocytes will support our observation that clinical trials of DI on DD patients are appropriate.

Referee #2 (Remarks for Author):

General remarks: This reports repurposes dantrolene, used to treat muscle spasticity or malignant hyperthermia, to treat the impaired endoplasmic reticulum calcium storage and defective cell-to-cell adhesion associated with Darier's skin

disease. The experiments are carefully done. More experimental detail could be added, as sometimes it is difficult to understand the experimental approach. In addition, the conclusion that clinical trials are the next step appears to be overstated, as dantrolene only appears to be effective in reversing functional defects only in low doses of thapsigargin.

Answer: Please see the previous answer (the comment is duplicated).

Specific Criticisms:

Methods:

1) For genetic analysis, were whole skin samples or samples containing only epidermis analyzed. If whole skin samples, why was dermis included since the pathology appears to be in the epidermis only?

Answer: We appreciate this point being raised by the reviewer. In the original experiment, we tested the extraction of RNA from epidermis only, dermis only, and epidermis + dermis and found that the RIN values from dermis samples were negligible (thus indicating that little dermis was included due to careful dissection). As such, in order to avoid altering the samples by lengthy enzymatic removal of the dermis from the epidermis, we reasoned that using epidermis + dermis samples was most appropriate. With regard to the reviewer's comments, we performed deconvolution analysis of the bulk RNAseq data through the xCell tool and found that there was negligible amounts of fibroblasts in the samples used, and that the main population of cells was keratinocytes (**Figure EV 1C**). In addition, through this analysis we were also able to determine that there were very few significant differences in cell type composition between lesional and non-lesional biopsies. Regarding Hailey-Hailey disease, please see our answer to point 3 below.

2) Cell culture: what was the calcium concentration in the cell culture medium?

Answer: We have answered this in the Methods section 'cell culture and treatments'.

3) Calcium measurements: please include what solutions were used for calcium measurements. If media was used as the solution, were the solutions gassed with appropriate CO₂?

Answer: Culture media was used and yes, it was gassed with appropriate CO₂. As mentioned in response to point 2 from reviewer 1, we have included more extensive details about the experimental setup and conditions used in the calcium measurement experiments in the 'Ca²⁺ imaging' section in the methods section, including details about the gas conditions during imaging.

Fig 2. Its difficult to understand exactly what happened in this figure without going back to the methods. Please add this information in the figure legend.

Answer: We agree with the point raised, and as such, have included a graphical cartoon to the figure to summarise the experimental details (Figure 2A). In addition, we have included more detail in the results section.

Figure 5. There appear to be variable responses for everything except CHOP. Explanation?

Answer: It appears that the rescue effect of Dantrolene on ER stress is mostly through CHOP in the keratinocytes, which might explain the reduction in apoptosis with dantrolene. The rest of the UPR is largely unaffected in keratinocytes however in patient fibroblasts overall UPR is reduced with dantrolene treatment. We elaborate on this in the Discussion section on UPR. Furthermore, please note that the ER stress figure now is Figure 4 and that we have added data on patient derived keratinocytes.

Figure 6. Dantrolene appears to be effective only at TG 10 nM dose for cell viability. Do the authors know what its effect is on ER calcium in micromolar concentrations? Figure 2 only shows a normalized fluorescence intensity.

Answer: No, measured calcium in relative units and therefore we do not know the concentrations in micromolar, but we substantially expanded our calcium measurements by analysis of cytosolic calcium dynamics using the ratiometric dye Fura Red AM, as well as mitochondrial calcium through live cell imaging of Rhod-2 AM (**Figure 2**). These experiments were also expanded to patient derived keratinocytes. Furthermore, regarding cell viability and thapsigargin concentrations, cell viability is a less sensitive assay than apoptosis which did show dantrolene rescue at high thapsigargin concentrations (**Figure 5F**).

Discussion: the authors state that there have been no clinical trials for Darier's disease. It's more accurate to say there have been no successful trials. Also it's worth discussing whether the changes in adhesion and apoptosis are large enough in cell culture to warrant a clinical trial.

Answer: We acknowledge the reviewers first comment and have addressed this to say that there have been no successful clinical trials (Discussion). With regards to the second point, as there are no validated animal models for DD, we feel that our statement about the need for clinical trials with DI is warranted in the face of our *in vitro* data. Moreover, such a trial would be based on drug repurposing of a safe drug that has been in use since the 1970's.

Referee #3 (Comments on Novelty/Model System for Author):

The authors chose to use siRNA knockdown of ATP2A2 in primary keratinocytes despite ready access to darier patients - knockdown is not the same as at the haploinsufficiency seen in DD patients. It's not clear that the model is true to the disease state. Furthermore, use of DD fibroblasts is odd given the absence of clear pathology in these cells in human disease.

Answer: First, thanks for your thorough review. We have addressed this by performing experiments on patient derived primary keratinocytes (**Figure 2-5**).

Referee #3 (Remarks for Author):

The authors chose to use siRNA knockdown of ATP2A2 in primary keratinocytes despite ready access to darier patients - knockdown is not the same as at the haploinsufficiency seen in DD patients. It's not clear that the model is true to the disease state. Furthermore, use of DD fibroblasts is odd given the absence of clear pathology in these cells in human disease. This data is not a surrogate for studying patient keratinocytes.

It is not surprising that a calcium channel inhibitor mitigates the effects of decreased ER calcium stores in DD..

The authors should cite manuscripts showing the IL17 inhibition has efficacy in DDD.

Answer: We thank the reviewer for their time reviewing our manuscript. With regards to their first point, we acknowledge that KD of *ATP2A2* is not the same as DD, and so have corrected this throughout the paper to state that the si*ATP2A2* HEKa cells are models of genetic SERCA2 inhibition. For the second point, whilst we agree with the point about DD fibroblasts, we feel that their inclusion is of some interest, due to the fact that DD pathogenesis affects several cell types and organs, although clearly not the same extent as keratinocytes. With regard to this, we have repeated essential experiments in primary DD keratinocytes (**Figure 2-5**) and kept the DD fibroblast data, although placed less importance to it. Regarding the effect of a calcium channel inhibitor, we agree that this is logical, and this is exactly why we chose to perform this study. It is a bit surprising that this has not been tested before, but bear in mind that Darier disease is quite rare and only studied by a few researchers. Finally, we agree with the suggestion about IL17 inhibition and have included text discussing this in the Discussion.

Figure 2: The authors appear to be seeing calcium retention in excess of normal keratinocytes (siNEG/DL), nearly 2-fold greater. Normalization would be expected to be more relevant. Panels C and D are from non-relevant cells and have expected results. E appear to have a typo on right upper panel. While confocal is interesting - graphical presentation of analysis of multiple cells would be of greater benefit. F - it is not clear why fibroblast data is presented.

Answer: Thank you for the comment. With regards to the comment about calcium retention in the siNEG/DI cells, calcium FI only goes up around 20% compared to siNEG/DI- cells, far less compared to siATP2A2/DI. This data has been moved the extended view (**EV2**) as we have added new data on patient derived primary keratinocytes that are more pathophysiologically relevant. We acknowledge that there may have been a misunderstanding in the experimental setup due to lack of details on our end. To this regard, we have specified that in the siATP2A2 cell model we did not use acute Tg treatment as we did in the DDK and Tg-HEKa models and instead used acute DI treatment. Finally, as suggested, we have added single cell calcium traces (**EV2 F-K**).

Given suggestion of therapeutic effect, have the authors considered RNA sequencing of normal keratinocytes, their *atp2a2* knockdown cells, and the same cells treated with DL to show global normalization of gene function.

**Karolinska
Institutet**

Answer: That is a good suggestion, however because of other reviewers' and the editor's request of primary patient keratinocytes, we had to focus our resubmission experiments there.

Overall, my greatest concern is the use of a non-physiologic model for DD. While the findings presented may be true, it is equally possible that they do not reproduce disease features of DD.

Answer: We acknowledge the reviewers preconceptions about the use of DD fibroblasts and as such have repeated many essential experiments in primary DDK keratinocytes (Figures 2-5).

4th Mar 2024

Dear Prof. Wikstrom,

Thank you for the submission of your revised manuscript to EMBO Molecular Medicine. We have now heard back from one referee who agreed to re-evaluate your manuscript. This referee also assessed your responses to concerns raised by referee #3. As you will see from the report below, the referee acknowledges the improvements of the revised manuscript but remains critical particularly regarding the calcium imaging and quantification. After a consultation with the referee and with my colleagues here, we agreed that raised concerns are justified and should be addressed in an additional and final round of major revision. We agreed that performing ratiometric measurements and calculations of calcium concentration are essential for further consideration of the manuscript. Other points incl. STIM1-gated ORAI channels and mitochondrial involvement should be addressed by discussion. Also, we agree with the referee that the manuscript would be better suited as a Report. Therefore, please check "Author Guidelines" and format your manuscript as a report.
<https://www.embopress.org/page/journal/17574684/authorguide#reportsarticleguide>

Please also amend following points:

- Please address all comments suggested by our data editors listed below:

o Figure legends:

1. Please note that the legends for figures 3i-j is not provided in the sequential manner (legend for figure 3j is provided before legend of figure 3i). This needs to be rectified.
 2. Please define the annotated p value * in the legends of figures 3i, k; as appropriate.
 3. Please indicate the statistical test used for data analysis in the legends of figures EV 1b-c.
 4. Please note that in figures 2c; 4a-d; there is a mismatch between the annotated p values in the figure legend and the annotated p values in the figure file that should be corrected.
 5. Please note that information related to n is missing in the legends of figures 3i, k.
 6. Please note that the error bars are not defined in the legends of figures EV 2b-c; EV 3a-b.
 7. Please note that scale bar and its definition are missing for figure 1a.
- Please combine Appendix Figure S1 and S3 and their legends in a single PDF file named Appendix with a table of content on the title page and page numbers. Appendix Figure S2 is a dataset and should be renamed and uploaded as such with the callout updated in the manuscript accordingly.
- Rename "Conflict-of-interest statement" to "Disclosure and competing interests statement". We updated our journal's competing interests policy in January 2022 and request authors to consider both actual and perceived competing interests. Please review the policy <https://www.embopress.org/competing-interests> and update your competing interests if necessary.

Further consideration of a revision that addresses reviewer's concerns in full will entail an additional round of review. Acceptance or rejection of the manuscript will depend on the completeness of your responses included in the next, final version of the manuscript. For this reason, and to save you from any frustrations in the end, I would strongly advise against returning an incomplete revision.

We would welcome the submission of a revised version within three months for further consideration. Please let us know if you require longer to complete the revision.

I look forward to receiving your revised manuscript.

Yours sincerely,

Zeljko Durdevic

We require:

2) Individual production quality figure files as .eps, .tif, .jpg (one file per figure). For guidance, download the 'Figure Guide PDF': (<https://www.embopress.org/page/journal/17574684/authorguide#figureformat>).

3) A .docx formatted letter INCLUDING the reviewers' reports and your detailed point-by-point responses to their comments. As part of the EMBO Press transparent editorial process, the point-by-point response is part of the Review Process File (RPF), which will be published alongside your paper.

4) A complete author checklist, which you can download from our author guidelines (<https://www.embopress.org/page/journal/17574684/authorguide#submissionofrevisions>). Please insert information in the checklist that is also reflected in the manuscript. The completed author checklist will also be part of the RPF.

6) It is mandatory to include a 'Data Availability' section after the Materials and Methods. Before submitting your revision, primary datasets produced in this study need to be deposited in an appropriate public database, and the accession numbers and database listed under 'Data Availability'. Please remember to provide a reviewer password if the datasets are not yet public (see <https://www.embopress.org/page/journal/17574684/authorguide#dataavailability>).

13) Author contributions: You will be asked to provide CRediT (Contributor Role Taxonomy) terms in the submission system. These replace a narrative author contribution section in the manuscript.

14) A Conflict of Interest statement should be provided in the main text.

Please also suggest a striking image or visual abstract to illustrate your article as a PNG file 550 px wide x 300-800 px high.

***** Reviewer's comments *****

Referee #1 (Remarks for Author):

The authors performed additional experiments and thus addressed some of the comments from the three reviewers. Nevertheless, some questions remain unanswered. These are described below and some are of essential importance for the claims of this paper.

Compared to other paper published in EMBO Mol Med the amount of data presented here is not on the higher end. Especially with the current figure organization (see for example Fig. 1 with only 3 panels) it would be, in my opinion, relatively easy to present the study as a report.

Regarding my comment about calcium imaging. I am somewhat puzzled as to why the authors used Fura Red and Rhod-2 for their cytosolic and mitochondrial measurements, respectively. These dyes do not allow ratiometric measurements and hence, calculation of the calcium concentration. My suggestion that normalizing to 1 is not appropriate for this study has been ignored. Without proper calibration of the calcium signals in all three compartments the claims of this paper are not fully supported by the data.

24 hours with Tg is more than sufficient to induce ER stress and Fig. 4B indicates that. I am not sure if I understand the explanation correctly. Do the authors suggest that ER stress has no functional role in this experimental system?

The new NFAT data indicate that indeed the cytosolic calcium is essential and obviously altered in DI-treated cells and mutant DDK patient cells. This again indicates that proper measurements and precise calibration of intracellular calcium are essential for this study. The authors did not comment on the contribution of STIM1-gated ORAI channels. As mentioned in my first review, STIM1 is proposed as a biomarker of DD. Is dantrolene affecting their activity?

The GO term analysis indicated significant mitochondrial involvement. I understand the concern of the authors that analyzing mitochondrial function requires significant work. Nevertheless, to better understand DD development and therapy a more comprehensive evaluation of the whole cellular reprogramming is essential. Furthermore, mitochondria are known to control ER Ca²⁺ store depletion as well as ORAI channel function.

Dear Prof. Wikstrom,

Thank you for the submission of your revised manuscript to EMBO Molecular Medicine. We have now heard back from one referee who agreed to re-evaluate your manuscript. This referee also assessed your responses to concerns raised by referee #3. As you will see from the report below, the referee acknowledges the improvements of the revised manuscript but remains critical particularly regarding the calcium imaging and quantification. After a consultation with the referee and with my colleagues here, we agreed that raised concerns are justified and should be addressed in an additional and final round of major revision. We agreed that performing ratiometric measurements and calculations of calcium concentration are essential for further consideration of the manuscript. Other points incl. STIM1-gated ORAI channels and mitochondrial involvement should be addressed by discussion. Also, we agree with the referee that the manuscript would be better suited as a Report. Therefore, please check "Author Guidelines" and format your manuscript as a report. <https://www.embopress.org/page/journal/17574684/authorguide#reportsarticleguide>

Answer: We have now performed ratiometric measurements and calculations of calcium concentrations (Figure 1D-K).

Please also amend following points:

- Please address all comments suggested by our data editors listed below:
o Figure legends:

1. Please note that the legends for figures 3i-j is not provided in the sequential manner (legend for figure 3j is provided before legend of figure 3i). This needs to be rectified.

Answer: We have corrected the sequential manner of call outs in the figure legends following reorganisation into a report.

2. Please define the annotated p value * in the legends of figures 3i, k; as appropriate.

Answer: We have corrected this for the now Figure 2I, K.

3. Please indicate the statistical test used for data analysis in the legends of figures EV 1b-c.

Answer: This has now been corrected.

4. Please note that in figures 2c; 4a-d; there is a mismatch between the annotated p values in the figure legend and the annotated p values in the figure file that should be corrected.

Answer: This has been corrected for the now Figure 1J and Figure 3A-D

5. Please note that information related to n is missing in the legends of figures 3i, k.

Answer: This has been updated for the new Figure 2I and 2K

6. Please note that the error bars are not defined in the legends of figures EV 2b-c; EV 3a-b.

Answer: We have added the error bar definitions to Figures EV3A-B

7. Please note that scale bar and its definition are missing for figure 1a.

Answer: Scale bar has been added to Figure 1A

- Please combine Appendix Figure S1 and S3 and their legends in a single PDF file named Appendix with a table of content on the title page and page numbers. Appendix Figure S2 is a dataset and should be renamed and uploaded as such with the callout updated in the manuscript accordingly.

Answer: We have now combined all appendix figures into a single PDF titled 'Appendix', and referred to call-outs within the manuscript.

- Rename "Conflict-of-interest statemen" to "Disclosure and competing interests statement". We updated our journal's competing interests policy in January 2022 and request authors to consider both actual and perceived competing interests. Please review the policy <https://www.embopress.org/competing-interests> and update your competing interests if necessary.

Answer: This has been updated on the title page.

Referee #1 (Remarks for Author):

The authors performed additional experiments and thus addressed some of the comments from the three reviewers. Nevertheless, some questions remain unanswered. These are described below and some are of essential importance for the claims of this paper.

Compared to other paper published in EMBO Mol Med the amount of data presented here is not on the higher end. Especially with the current figure organization (see for example Fig. 1 with only 3 panels) it would be, in my opinion, relatively easy to present the study as a report.

Regarding my comment about calcium imaging. I am somewhat puzzled as to why the authors used Fura Red and Rhod-2 for their cytosolic and mitochondrial measurements, respectively. These dyes do not allow ratiometric measurements and hence, calculation of the calcium concentration. My suggestion that normalizing to 1 is not appropriate for this study has been ignored.

Without proper calibration of the calcium signals in all three compartments the claims of this paper are not fully supported by the data.

Answer: That you for taking the time to review our manuscript again as well as your comment. With regards to this point, Fura Red is a ratiometric cytosolic calcium indicator, and so we have now performed additional experiments calculating the calcium concentration. As the mitochondrial calcium experiments were peripheral to the aim of the paper, as well as the fact that Rhod-2 is not ratiometric, we have not included that in the revised version, and have instead focussed on the Fura Red experiments.

24 hours with Tg is more than sufficient to induce ER stress and Fig. 4B indicates that. I am not sure if I understand the explanation correctly. Do the authors suggest that ER stress has no functional role in this experimental system?

Answer: Thanks for your comment. We treated normal primary keratinocytes (HEKa) with low levels of Tg for 24 hours in order to induce a pharmacological model of SERCA2 inhibition, in concert with the primary DD keratinocytes and si KD model. In the ER stress experiment that you refer to we used Tg pretreatment and then investigated whether dantrolene had a rescue effect.

The new NFAT data indicate that indeed the cytosolic calcium is essential and obviously altered in DI-treated cells and mutant DDK patient cells. This again indicates that proper measurements and precise calibration of intracellular calcium are essential for this study. The authors did not comment on the contribution of STIM1-gated ORAI channels. As

**Karolinska
Institutet**

mentioned in my first review, STIM1 is proposed as a biomarker of DD. Is dantrolene affecting their activity?

Answer: Thank you for the comment. You raise a valid point. As we have reformatted as a report, we were not able to include further experiments looking into the role of STIM-gated ORAI channels, although we have however mentioned the potential importance in the relevant results/discussion section.

The GO term analysis indicated significant mitochondrial involvement. I understand the concern of the authors that analyzing mitochondrial function requires significant work. Nevertheless, to better understand DD development and therapy a more comprehensive evaluation of the whole cellular reprogramming is essential. Furthermore, mitochondria are known to control ER Ca²⁺ store depletion as well as ORAI channel function.

Answer: Again, this is another valid point, but unfortunately as we have reformatted this manuscript into a report, we have been unable to include additional experiments to investigate mitochondrial involvement. We have again however mentioned this in the relevant results/discussion section.

Dear Prof. Wikstrom,

Thank you for the submission of your revised manuscript to EMBO Molecular Medicine. We have now heard back from one referee who agreed to re-evaluate your manuscript. This referee also assessed your responses to concerns raised by referee #3. As you will see from the report below, the referee acknowledges the improvements of the revised manuscript but remains critical particularly regarding the calcium imaging and quantification.

After a consultation with the referee and with my colleagues here, we agreed that raised concerns are justified and should be addressed in an additional and final round of major revision. We agreed that performing ratiometric measurements and calculations of calcium concentration are essential for further consideration of the manuscript. Other points incl. STIM1-gated ORAI channels and mitochondrial involvement should be addressed by discussion. Also, we agree with the referee that the manuscript would be better suited as a Report. Therefore, please check "Author Guidelines" and format your manuscript as a

report. <https://www.embopress.org/page/journal/17574684/authorguide#reportsarticleguide>

Answer: It is formatted as a report.

Please also amend following points:

- Please address all comments suggested by our data editors listed below:
o Figure legends:

1. Please note that the legends for figures 3i-j is not provided in the sequential manner (legend for figure 3j is provided before legend of figure 3i). This needs to be rectified.

Answer: We have corrected the sequential manner of call outs in the figure legends following reorganisation into a report.

2. Please define the annotated p value * in the legends of figures 3i, k; as appropriate.

Answer: We have corrected this for the now Figure 2I, K

3. Please indicate the statistical test used for data analysis in the legends of figures EV 1b-c.

Answer: This has now been corrected.

4. Please note that in figures 2c; 4a-d; there is a mismatch between the annotated p values in the figure legend and the annotated p values in the figure file that should be corrected.

Answer: This has been corrected for the now Figure 1J and Figure 3A-D

5. Please note that information related to n is missing in the legends of figures 3i, k.

Answer: This has been updated for the new Figure 2I and 2K

6. Please note that the error bars are not defined in the legends of figures EV 2b-c; EV 3a-b.

Answer: We have added the error bar definitions to Figures EV3A-B

7. Please note that scale bar and its definition are missing for figure 1a.

Answer: Scale bar has been added to Figure 1A

- Please combine Appendix Figure S1 and S3 and their legends in a single PDF file named Appendix with a table of content on the title page and page numbers. Appendix Figure S2 is a dataset and should be renamed

**Karolinska
Institutet**

and uploaded as such with the callout updated in the manuscript accordingly.

Answer: We have now combined all appendix figures into a single PDF titled 'Appendix', and referred to call-outs within the manuscript.

- Rename "Conflict-of-interest statemen" to "Disclosure and competing interests statement". We updated our journal's competing interests policy in January 2022 and request authors to consider both actual and perceived competing interests. Please review the policy <https://www.embopress.org/competing-interests> and update your competing interests if necessary.

Answer: This has been updated on the title page.

7th Mar 2024

Dear Prof. Wikstrom,

Thank you for your response to the editorial decision on your manuscript entitled "Dantrolene corrects cellular disease features of Darier disease and may be a novel treatment". I have now carefully examined the arguments provided in your letter and discussed them with the other members of our editorial team. Additionally, I have sought external advice on the study from an expert in the field.

I regret to inform you that we will not be able to reverse our original decision. We do acknowledge the improvements of the revised manuscript in particular the experiments with primary keratinocytes; however, in line with our editorial assessment our adviser agreed with the concern raised by the referee #1 regarding calcium imaging and quantification. Below are the comments from our adviser for your consideration.

- The reviewer asks to use ratiometric Ca²⁺ indicators, which is a valid request. The authors use FuraRed, which is indeed ratiometric, but do not convert the fluorescence value in [Ca²⁺]. This can be done by performing the usual calibration with cell permeabilization/Ca²⁺ supplementation/Ca²⁺ chelation. It is standard practice to do so. In my opinion, normalization of FuraRed shall be done.
- Along the same line, I don't understand how FuraRed was imaged. The details in the methods section are scant, with a reference to the in-built filters of the microscope, but common practice is to give details about the measurement because this dye is used in ratiometric measurements using excitation wavelengths of 420 nm and 480 nm or 457 nm and 488 nm. It is advisable that the authors specify that they used the 457-488 lasers and the bandwidth of the emission filter used to collect the images, as well as the procedure for ratioing and calculating the FuraRed ratio. These details appear to be pedantic but are essential to interpret the Ca²⁺ measurements carried out by the authors.
- The rhod2 indicator is non-ratiometric and has a high K_d, meaning that it is saturated at mito [Ca²⁺] of 1 microM or less, way below the actual Ca²⁺ concentrations reported to occur in mitochondria especially upon IP₃-mediated Ca²⁺ signaling. The picture is however different when Ca²⁺ discharge from the ER is induced by SERCA inhibition with TG, as authors do here. Under these circumstances, mitochondrial Ca²⁺ uptake is much reduced and therefore Rhod2 might work. However, it is clear from the traces presented by the authors that Rhod2 is readily saturated also in their experimental conditions, as evidenced by the fact that DI and control treated cells reach the same peak in mitochondrial Ca²⁺ concentrations upon TG treatment. In any case, this experiment is peripheral to the message of the paper, unless authors want to say that less ER Ca²⁺ in DI treated cells leads to lower mitochondrial Ca²⁺ overload, which must be measured in completely different experiments.
- Did the authors pretreat cells with ionomycin before measuring Rhod2? This is unclear from the methods, and is clearly saturating the signal, so these measurements can be uninterpretable.
- Unclear if Ca²⁺ experiments were performed in Ca²⁺ free media (as advisable).
- These can all be corrected during revision as they are doable in 2 months. I would suggest that the authors go for another round of revision and perform additional Ca²⁺ experiments.

At this point, I would also like to clarify that we have invited all three initial referees to re-review your revised manuscript; however, only referee #1 accepted and was therefore asked to assess your responses to concerns of other referees, who declined to re-evaluate the manuscript.

Taking our adviser's comments in consideration and together with the referee re-evaluation and our editorial assessment we would like to reiterate our invitation of the second round of major revision. We would welcome the submission of a revised version within three months for further consideration. Please let us know if you require longer to complete the revision.

Please use this link to login to the manuscript system and submit your revision: <https://embomolmed.msubmit.net/cgi-bin/main.plex>

I look forward to receiving your revised manuscript.

Yours sincerely,

Zeljko Durdevic

We require:

2) Individual production quality figure files as .eps, .tif, .jpg (one file per figure). For guidance, download the 'Figure Guide PDF': (<https://www.embopress.org/page/journal/17574684/authorguide#figureformat>).

3) A .docx formatted letter INCLUDING the reviewers' reports and your detailed point-by-point responses to their comments. As part of the EMBO Press transparent editorial process, the point-by-point response is part of the Review Process File (RPF), which will be published alongside your paper.

4) A complete author checklist, which you can download from our author guidelines (<https://www.embopress.org/page/journal/17574684/authorguide#submissionofrevisions>). Please insert information in the checklist that is also reflected in the manuscript. The completed author checklist will also be part of the RPF.

6) It is mandatory to include a 'Data Availability' section after the Materials and Methods. Before submitting your revision, primary datasets produced in this study need to be deposited in an appropriate public database, and the accession numbers and database listed under 'Data Availability'. Please remember to provide a reviewer password if the datasets are not yet public (see <https://www.embopress.org/page/journal/17574684/authorguide#dataavailability>).

13) Author contributions: You will be asked to provide CRediT (Contributor Role Taxonomy) terms in the submission system. These replace a narrative author contribution section in the manuscript.

14) A Conflict of Interest statement should be provided in the main text.

Please also suggest a striking image or visual abstract to illustrate your article as a PNG file 550 px wide x 300-800 px high.

Dear Prof. Wikstrom,

Thank you for your response to the editorial decision on your manuscript entitled "Dantrolene corrects cellular disease features of Darier disease and may be a novel treatment". I have now carefully examined the arguments provided in your letter and discussed them with the other members of our editorial team. Additionally, I have sought external advice on the study from an expert in the field.

I regret to inform you that we will not be able to reverse our original decision. We do acknowledge the improvements of the revised manuscript in particular the experiments with primary keratinocytes; however, in line with our editorial assessment our adviser agreed with the concern raised by the referee #1 regarding calcium imaging and quantification. Below are the comments from our adviser for your consideration.

- The reviewer asks to use ratiometric Ca^{2+} indicators, which is a valid request. The authors use FuraRed, which is indeed ratiometric, but do not convert the fluorescence value in $[\text{Ca}^{2+}]$. This can be done by performing the usual calibration with cell permeabilization/ Ca^{2+} supplementation/ Ca^{2+} chelation. It is standard practice to do so. In my opinion, normalization of FuraRed shall be done.
- Along the same line, I don't understand how FuraRed was imaged. The details in the methods section are scant, with a reference to the in-built filters of the microscope, but common practice is to give details about the measurement because this dye is used in ratiometric measurements using excitation wavelengths of 420 nm and 480 nm or 457 nm and 488 nm. It is advisable that the authors specify that they used the 457-488 lasers and the bandwidth of the emission filter used to collect the images, as well as the procedure for ratioing and calculating the FuraRed ratio. These details appear to be pedantic but are essential to interpret the Ca^{2+} measurements carried out by the authors.

Answer: Thank you for the helpful comment. We have now clarified the specific experimental conditions in which we performed the Fura Red imaging and calculations within the methods section.

- The rhod2 indicator is non-ratiometric and has a high K_d , meaning that it is saturated at mito $[\text{Ca}^{2+}]$ of 1 μM or less, way below the actual Ca^{2+} concentrations reported to occur in mitochondria especially upon IP_3 -mediated Ca^{2+} signaling. The picture is however different when Ca^{2+} discharge from the ER is induced by SERCA inhibition with TG, as authors do here. Under these circumstances, mitochondrial Ca^{2+} uptake is much reduced and therefore Rhod2 might work. However, it is clear from the

traces presented by the authors that Rhod2 is readily saturated also in their experimental conditions, as evidenced by the fact that DI and control treated cells reach the same peak in mitochondrial Ca²⁺ concentrations upon TG treatment. In any case, this experiment is peripheral to the message of the paper, unless authors want to say that less ER Ca²⁺ in DI treated cells leads to lower mitochondrial Ca²⁺ overload, which must be measured in completely different experiments.

- Did the authors pretreat cells with ionomycin before measuring Rhod2? This is unclear from the methods, and is clearly saturating the signal, so these measurements can be uninterpretable.

- Unclear if Ca²⁺ experiments were performed in Ca²⁺ free media (as advisable).

- These can all be corrected during revision as they are doable in 2 months. I would suggest that the authors go for another round of revision and perform additional Ca²⁺ experiments.

Answer: Your in-depth comments and advice are much appreciated. Taking on board your advice, we have now removed the Rhod-2 mitochondrial calcium experiments from our manuscript and instead discussed the potential role of mitochondrial calcium in the relevant results/discussion section.

4th Jul 2024

Dear Prof. Wikstrom,

Please find enclosed the final reports on your manuscript. We are pleased to inform you that your manuscript is accepted for publication and is now being sent to our publisher to be included in the next available issue of EMBO Molecular Medicine.

Referee #4 (Remarks for Author):

I have been called to evaluate the Ca²⁺ experiments at the last round of revision. I made specific requests on the use of Rhod2 and on the calibration of FuraRed. Authors performed the Fura Red calibration and decided to eliminate the mitochondrial Ca²⁺ measurements using Rhod2 because of the intrinsic problems with that probe. They also clarified methods for Ca²⁺ measurements. I therefore believe that they have addressed my concerns, at least partially, and that therefore I have no other outstanding issues with the Ca²⁺ measurements part of the paper.
